# Two new multirotor Uncrewed Aerial Vehicles (UAVs) for glaciogenic cloud seeding and aerosol measurements within the CLOUDLAB project

Anna J. Miller[1], Fabiola Ramelli[1], Christopher Fuchs[1], Nadja Omanovic[1], Robert Spirig[1], Huiying Zhang[1], Ulrike Lohmann[1], Zamin A. Kanji[1], and Jan Henneberger[1]

[1]Institute for Atmospheric and Climate Science, ETH Zurich, Zurich, 8092 Switzerland

**Correspondence:** Anna J. Miller (anna.miller@env.ethz.ch) and Jan Henneberger (jan.henneberger@env.ethz.ch)

**Abstract.** Uncrewed Aerial Vehicles (UAVs) have become widely used in a range of atmospheric science research applications. Because of their small size, flexible range of motion, adaptability, and low cost, multirotor UAVs are especially well-suited for probing the lower atmosphere. However, their use so far has been limited to conditions outside of clouds, first because of the difficulty of flying beyond visual line of sight, and second because of the challenge of flying in icing conditions in supercooled clouds. Here, we present two UAVs for cloud microphysical research: one UAV (the measurement UAV) equipped with a Portable Optical Particle Spectrometer (POPS) and meteorological sensors to probe the aerosol and meteorological properties in the boundary layer, and one UAV (the seeding UAV) equipped with seeding flares to produce a plume of particles that can nucleate ice in supercooled clouds. A propeller heating mechanism on both UAVs allows for operating in supercooled clouds with icing conditions. These UAVs are an integral part of the CLOUDLAB project in which glaciogenic cloud seeding of supercooled low stratus clouds is utilized for studying aerosol-cloud interactions and ice crystal formation and growth.

In this paper, we first show validations of the POPS onboard the measurement UAV, demonstrating that the rotor turbulence has a small effect on measured particle number concentrations. We then exemplify the applicability for profiling the planetary boundary layer, as well as for sampling and characterizing aerosol plumes, in this case, the seeding plume. We also present a new method for filtering out high-concentration data to ensure good data quality of POPS. We explain the different flight patterns that are possible for both UAVs, namely horizontal or vertical leg patterns or hovering, with an extensive and flexible parameter space for designing the flight patterns according to our scientific goals. Finally, we show two examples of seeding experiments: first characterizing an out-of-cloud seeding plume with the measurement UAV flying horizontal transects through the plume, and second, characterizing an in-cloud seeding plume with downstream measurements from a POPS and a holographic imager mounted on a tethered balloon. Particle number concentrations and particle number size distributions of the seeding plume from the experiments reveal that we can successfully produce and measure the seeding plume, both in-cloud (with accompanying elevated ice crystal number concentrations) and out-of-cloud. The methods presented here will be useful for probing the lower atmosphere, for characterizing aerosol plumes, and for deepening our cloud microphysical understanding through cloud seeding experiments, all of which have the potential to benefit the atmospheric science community.

# 1 Introduction

In situ measurements of the atmosphere, and especially of clouds, are important for understanding and predicting Earth's weather and climate, especially for the energy balance, air quality, hydrological cycle, and other applications. In situ data can complement remote sensing measurements and are also used for initializing and validating weather prediction models important for our daily lives, for example, precipitation forecasts. However, obtaining in situ atmospheric measurements can be challenging, especially in the lower troposphere and in clouds. Uncrewed Aerial Vehicles (UAVs) present one major solution

to the challenge of probing the lower troposphere, filling the gap between ground-based and high-altitude measurements. Because UAVs are typically small, cost-efficient, reusable, and adaptable for a range of purposes, they can be an excellent addition to the more traditional in situ atmospheric measurement systems like weather balloons and crewed aircraft. Indeed, in recent years UAVs have been increasingly deployed for such purposes. For example, by installing a lightweight optical particle counter or particulate matter sensor, UAVs are well-suited for measuring vertical and/or horizontal distribution of aerosol in

the polluted boundary layer (e.g., Weber et al., 2017; Mamali et al., 2018; Samad et al., 2022; Li et al., 2022; Suchanek et al., 2022; Pusfitasari et al., 2023; Järvi et al., 2023). Other examples of UAVs used in atmospheric research include estimating atmospheric turbulence (e.g., Fuertes et al., 2019; Alaoui-Sosse et al., 2019; Egerer et al., 2023), measuring volcanic plumes and their dispersions (e.g., McGonigle et al., 2008; Mori et al., 2016; Albadra et al., 2020), and for meteorological profiling (e.g., Holland et al., 2001; Reuder et al., 2009; Brosy et al., 2017; Koch et al., 2018; Leuenberger et al., 2020; Brus et al., 2021;

Bärfuss et al., 2023).

For probing clouds, however, UAVs traditionally face challenges. First, because clouds hinder visibility, it is impossible to fly within visual line of sight into a cloud, and obtaining permission to fly beyond visual line of sight can be difficult due to regulatory frameworks. Second, like conventional crewed aircraft, UAVs can experience significant ice buildup in supercooled clouds, impacting flight performance or leading to a crash. Icing can occur at temperatures below 0 °C and depends on factors

such as temperature, liquid water content, ice water content, and cloud droplet size distributions (Bernstein et al., 2005). Ice buildup can occur very quickly on the propellers of a UAV such that the UAV cannot sustain its position and could fly off track or crash down, faster than the pilot can control or prevent it (Catry et al., 2021; Müller et al., 2023). However, one solution to the icing problem on multirotor UAVs is to install heated propellers which can prevent ice from building up, as has been developed for the Meteodrone® (Meteomatics AG, 2023). With these Meteodrones, we were able to develop a unique method

for in situ glaciogenic cloud seeding and downwind aerosol measurements, even in severe icing conditions.

Glaciogenic cloud seeding is the process of injecting substances into supercooled clouds to initiate primary ice formation. Ice in clouds is important for the atmosphere and climate for several reasons, namely because most continental precipitation is formed via the ice phase (Mülmenstädt et al., 2015; Heymsfield et al., 2020) and because ice crystals affect the radiative properties and lifetime of clouds. Primary ice forms in clouds through two pathways: homogeneous nucleation, where supercooled

water spontaneously freezes, or heterogeneous nucleation, where an ice nucleating particle (INP) gives the supercooled water a surface to freeze onto, thereby lowering the energy barrier to ice nucleation (Kanji et al., 2017; Knopf and Alpert, 2023). Homogeneous ice nucleation can only occur when cloud droplets are supercooled to below -38 °C, whereas heterogeneous

nucleation occurs at warmer temperatures, even up to -1 ° C, depending on the seed particle type and size (Kanji et al., 2017). In glaciogenic cloud seeding, the heterogeneous ice nucleation process is exploited: particles that are effective INPs (e.g., silver iodide) are injected into supercooled clouds to artificially initiate ice crystal formation (Dennis, 1980; Rauber et al., 2019). Once the ice crystals form, they grow by vapor deposition and collisions, and may grow large enough to precipitate from the cloud. Therefore, there is interest in cloud seeding as a tool for weather modification but also as a tool for developing our scientific understanding of ice evolution in supercooled mixed-phase clouds.

The first glaciogenic cloud seeding experiments were conducted in the 1940s by Schaefer (1946) using dry ice and Vonnegut (1947) using silver iodide particles, followed by a lot of operational cloud seeding activities in the 1970s intending to increase precipitation. However, mixed results of the effectiveness of these activities caused waning enthusiasm (see reviews of e.g., Dennis (1980) or Bruintjes (1999)). Currently, despite mixed evidence and continued debates about its efficacy (WMO, 2018; Rauber et al., 2019; Benjamini et al., 2023), there is a renewed interest in cloud seeding with operational seeding projects occurring across the world (e.g., Griffith et al., 2009; Woodley and Rosenfeld, 2004; Kulkarni et al., 2019; Wang et al., 2019; Al Hosari et al., 2021). Some studies, like the SNOWIE project (French et al., 2018; Friedrich et al., 2021), have a strong scientific component but are attached to operational seeding projects, limiting their experimental possibilities. Further, cloud seeding efforts are usually executed using either crewed aircraft or ground-based seeding techniques to disperse the INPs into clouds, but both pose constraints in terms of cost and flexibility – UAVs can provide a solution to these constraints. A few recent studies have presented methods for operational cloud seeding using fixed-wing UAVs (Jung et al., 2022; DeFelice et al., 2023), which have long flight times compared to multirotor UAVs, but with the sacrifice of precise control. Multirotor UAVs, therefore, are uniquely advantageous for cloud seeding from a scientific perspective, where precision and repeatability are necessary and large-scale seeding is not needed.

In our project named "CLOUDLAB", we use a multirotor UAV to seed persistent wintertime low stratus clouds as they allow for repeatable glaciogenic cloud seeding and laboratory-like adjustments of experimental parameters (e.g., seeding distance, which directly translates into ice crystal growth time) (Henneberger et al., 2023). Using a second multirotor UAV, we fly downstream of the seeding location to measure and monitor the seeding plume, while simultaneously measuring the cloud microphysical changes with other in situ and remote sensing instrumentation. Together, the seeding and downstream measurements can help us to better understand aerosol and cloud microphysical processes in mixed-phase clouds.

Here we present our novel method for glaciogenic cloud seeding and in situ atmospheric aerosol measurements with two modified, commercial, multirotor UAVs. The measurement UAV can measure particle number size distributions and particle number concentrations using an attached Portable Optical Particle Spectrometer (POPS), making the UAV well suited for atmospheric aerosol profiling as well as for measuring and characterizing the plume of seeding particles. The seeding UAV can burn up to two burn-in-place seeding flares while flying in a supercooled cloud with icing conditions, so it can effectively seed cloud regions with temperatures cold enough to glaciate. Both UAVs fly autonomously and have several distinct preprogrammed mission types with adjustable parameters for a range of experiment types (Sect. 2.4), allowing for a variety of flexible and targeted seeding and measurement missions. In the following, we present the technical and scientific capabilities of the measurement and seeding UAVs (Sect. 2), validation studies for the particle measurements with the measurement UAV

(Sect. 3), determination of the planetary boundary layer similar to radiosondes (Sect. 4), and the methods for in-cloud and out-of-cloud seeding experiments, with selected results from the first two CLOUDLAB campaigns (Sect. 5).

## 2 Instrumentation and field site description

### 2.1 Meteodrones

Both the measurement UAV and the seeding UAV are adapted Meteodrones (MM-670, Meteomatics AG, Switzerland), shown in Fig. 1. These Meteodrones are 6-rotor UAVs with a $70\,\mathrm{cm}$ diameter and a weight of $5\,\mathrm{kg}$, able to carry up to $1\,\mathrm{kg}$ of instrumental payload. They can fly for approximately 20 minutes at a maximum speed of $10\,\mathrm{m\,s^{-1}}$ and can withstand wind speeds up to $25\,\mathrm{m\,s^{-1}}$. They were developed to be used for frequent automatic atmospheric profiling up to $6\,\mathrm{km}$ above mean sea level (amsl) for the assimilation of their meteorological data into numerical weather prediction models (Leuenberger et al., 2020). The standard version of the Meteodrone is equipped with sensors to measure temperature ($\pm 0.1\,\mathrm{K}$; Integrated Circuit temperature sensor), relative humidity ($\pm 1.8\%$ at $23\,^\circ\mathrm{C}$ between 0-90% RH; capacitive sensor with humidity-permeable cover layer), and pressure ($\pm 1.5\,\mathrm{hPa}$; Piezo-resistive sensor), as well as a calibrated system for measuring wind speed ($\pm 1\,\mathrm{m\,s^{-1}}$) and wind direction ($\pm 10°$), each at $10\,\mathrm{Hz}$ sampling frequency (Meteomatics personal communication; Hervo et al., 2023). Meteorological measurements are post-processed by a Meteomatics algorithm to account for sensor calibrations and to combine the data from the ascent and descent flight of a vertical profile. All meteorological measurements are validated and calibrated by the manufacturer for the operational profiling flight speed of $10\,\mathrm{m\,s^{-1}}$.

The Meteodrone MM-670 model features integrated propeller heating to prevent ice from building up on the blades, allowing flights into supercooled clouds. An algorithm in the UAV controller software gives a warning when icing may be occurring according to the real-time UAV temperature and humidity data, but the propeller heating mechanism needs to be activated manually by the pilot. The pilot's decision to activate the propeller heating arises through a combination of assessing the algorithm warning output, the trend of the current battery consumption of the UAV, as well as knowledge and observations of the weather conditions the UAV is experiencing. Upon activation, the propeller heating turns on for 10 seconds. In intense icing conditions, the heating may be activated repeatedly for as long as it is deemed necessary (or until conditions are estimated to be too harsh and the flight is aborted). The downside of the electrothermal deicing mechanism is the high power consumption. Thus, there is a trade-off between the length of flight time and the amount of propeller heating needed, and pilots must be well-trained to handle icing situations appropriately to avoid potential damage or loss of the UAV.

Finally, the Meteodrones are also equipped with an emergency recovery system, including a parachute that is released automatically or on-demand in emergency situations, for example in the case of engine failure. The Meteodrone parachute system, as well as appropriate pilot training, allow us to obtain airspace permissions to be able to fly beyond visual line of sight in autonomous missions.

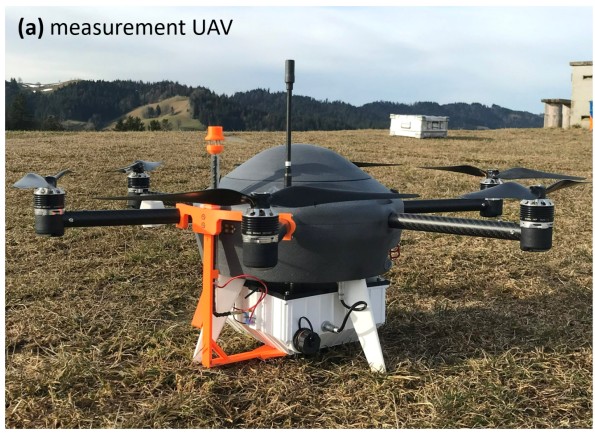 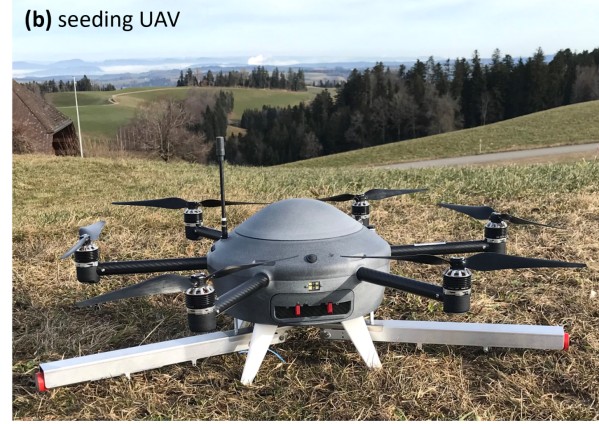

**Figure 1.** Images of the two UAVs: **(a)** the measurement UAV, a Meteodrone equipped with a Portable Optical Particle Spectrometer (white box) and an extended inlet (orange-capped tube), and **(b)** the seeding UAV, a Meteodrone with two attached burn-in-place seeding flares.

## 2.2 Measurement UAV

The measurement UAV (Fig. 1a) is equipped with a Portable Optical Particle Spectrometer (POPS, Handix Scientific, USA).
The POPS is a lightweight (550 g) optical particle counter measuring particle number and particle size distribution in the range of 115 nm - 3.37 μm at a 1-second time resolution, with a suggested flow rate of $3\,\mathrm{cm^3\,s^{-1}}$ (possible range of 0.083 to $5.83\,\mathrm{cm^3\,s^{-1}}$) (Handix Scientific, 2023). POPS was designed to be used on mobile platforms and has already been deployed with success on radiosonde balloons (Yu et al., 2017, 2019; Kloss et al., 2020), tethered balloons (de Boer et al., 2018; Creamean et al., 2021; Pilz et al., 2022; Mei et al., 2022; Walter et al., 2023; Lata et al., 2023), fixed-wing UAVs (Telg et al., 130 2017; Kezoudi et al., 2021; Mei et al., 2022; DeFelice et al., 2023), and other multirotor UAVs (Liu et al., 2021; Brus et al., 2021).

The POPS on the measurement UAV (referred to hereafter as POPS$_{\mathrm{UAV}}$) is attached to the bottom of the UAV with a custom, 3D-printed, water-tight housing. An inlet extension was designed so that the inlet (2 mm inner diameter, not isokinetic) extends out of the housing, bends 90° upwards, and extends up to 5 cm above the level of the rotors (the orange-capped tube in Figure 135 1a). Flow rates of $3\,\mathrm{cm^3\,s^{-1}}$ or $0.9\,\mathrm{cm^3\,s^{-1}}$ were used for POPS$_{\mathrm{UAV}}$. The inlet also includes a coiled heating wire to prevent the build-up of ice. The sampled particles are not dried prior to measurement, thus POPS$_{\mathrm{UAV}}$ reports particle diameters that are humidity-dependent and can be interpreted along with the relative humidity measured by the Meteodrone sensor. A detailed discussion of the inlet sampling efficiencies is given in Appendix A.

## 2.3 Seeding UAV

The seeding UAV (Fig. 1b) is modified to be able to ignite up to two burn-in-place seeding flares. Attached to the underside of the body of the UAV are two aluminum holders to host the flares. Flare ignition wires are connected to the UAV, and ignition is controlled by the UAV control software which ignites the flare with an electrical pulse at the predetermined ignition point

along the seeding pattern. A safety precaution is in place such that the flare will not ignite unless the drone is at least 105 m above ground. When the flare ignites, there is an audible sound, and if out-of-cloud, a visible plume (Fig. 2). The seeding flares we use (Zeus MK2, Cloud Seeding Technologies) consist of 200 g of material containing a mixture of silver iodide, silver chloride, ammonium salt, and potassium salt, of which around 20 g is ice-active material (Cloud Seeding Technologies, personal communication). One seeding flare burns for 5 - 6 minutes, and we have the option of using up to two flares simultaneously or consecutively.

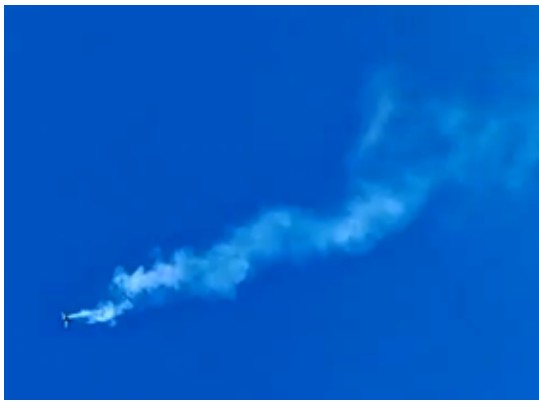

**Figure 2.** Image of the seeding UAV hovering at 150 m above ground while an attached flare is burning (out-of-cloud). The photo contrast was enhanced to make the plume more visible.

## 2.4  UAV flight patterns

The Meteodrone software was modified for us to be able to fly our desired measurement and seeding patterns. Both the measurement and seeding UAV can autonomously fly predefined flight patterns, and all patterns can be performed either in-cloud or out-of-cloud. The execution of the flight patterns is entirely autonomous, with the pilot only needing to program the flight mission using a set of mission-specific parameters, to "launch" the mission after completing a pre-flight checklist (e.g., checking weather, airspace clearance, the physical UAV itself, and its battery), as well as to activate deicing as needed.

The parameter space available to us in configuring a mission, where a "mission" is considered one complete flight by a UAV, is illustrated in Fig. 3. First, during the planning stage of a mission, we consider the prevailing environmental conditions using a combination of remote sensing and in situ measurements to decide on an appropriate altitude and location for seeding. The most important variables for our decision are wind speed, wind direction, cloud base altitude, cloud top altitude, temperature profile within the cloud, and cloud structure (i.e., cloud radar reflectivity). For example, when we plan an in-cloud seeding mission, during which we expect to nucleate and measure ice crystals, we target stable low stratus clouds with cloud temperatures below $-5°$ C (cold enough for ice nucleation to occur with silver iodide particles), low radar reflectivity (i.e., low background ice content), cloud base between 1100 and 1600 m amsl (low enough to be reached with our UAVs and tethered balloon), and

wind speeds of 3-15 m s$^{-1}$ (high enough to get advection of the seeding plume, and low enough to have safe conditions for flight of UAV and balloon).

After having determined the mission parameters, we use a custom-programmed website interface to calculate the seeding pattern start coordinates ($x_1$, $y_1$, and $z_1$ in Fig. 3) as well as the closest launch site, which is chosen from a set of pre-selected UAV launch locations surrounding our main measurement site (more details in Henneberger et al., 2023). Our mission configuration can be any number of horizontal or vertical legs ($n$), with any length of leg ($L$), and any horizontal distance between legs ($dx$), within our airspace allowance around our main site (see Section 2.5). Additionally, we can set the flight

speed of the UAV ($v_1$), the direction of the flight pattern ($\alpha$) and a waiting time after each leg ($t_{wait}$), which is useful in the case where we want the UAV to remain stationary while seeding (parameters $n = 1$, $L = 0$, and $dx = 0$ with $t_{wait} = 5$ minutes). Finally, there is a parameter to set whether to ignite the first seeding flare and if/when to ignite the second flare. The first seeding flare ignites (if set to do so) when the UAV reaches the pattern start point ($x_1$, $y_1$, $z_1$), while the second flare ignition point can be set to the start of a specified leg. The flight patterns and parameter space are used for designing the flight pattern

of both UAVs; all parameters are the same for a measurement mission except the flare ignition. Based on these parameters, we can flexibly design experiments to suit the current environmental conditions and our different scientific questions.

## 2.5   CLOUDLAB field site and other instrumentation

So far, the CLOUDLAB project has conducted two wintertime field measurement campaigns, in January 2022 - March 2022 and in December 2022 - February 2023, and a third campaign is planned for December 2023 - February 2024. The main

field site of the campaigns is in the central Swiss Plateau region in Eriswil, Switzerland (main site coordinates: 47°04'14"N, 7°52'22"E, 920 m elevation). We obtained air space clearance for our experiments with an area of a 4 km radius and a 2 km amsl height (1080 m above ground relative to the main site). At the main measurement site, we have a suite of in situ and ground-based remote sensing instrumentation, detailed in Henneberger et al. (2023).

    Remote sensing instruments relevant to the results presented here include: a ceilometer (CHM 15K, Lufft) for detecting cloud

base height and planetary boundary layer height, a cloud radar (Mira-35, Metek) for detecting cloud top and cloud structure including the seeding signal, and a radar wind profiler (LAP-3000, Vaisala) provided by MeteoSwiss for measuring vertically-resolved wind speeds and directions. Relevant in situ devices, besides the UAVs, are radiosondes (Sparv S1H3, Windsond) for obtaining vertical profiles of temperature, humidity, and wind, as well as a tethered balloon system (TBS). The measurement platform on the TBS (can be seen in Fig. 4b) is equipped with a holographic imager (HOLIMO) to measure characteristics

of cloud droplets and ice crystals with a size range of 6 μm – 2 mm (Ramelli et al., 2020) and a POPS (referred to hereafter as POPS$_{TBS}$) for measuring aerosol. The instrumentation aboard the TBS is deployed during in-cloud seeding experiments to measure the aerosol particles, cloud droplets, and ice crystals inside the seeding plume.

    POPS$_{TBS}$ has an inlet design identical to that of POPS$_{UAV}$ (see Section 2.2). A flow rate of 3 cm$^3$ s$^{-1}$ was used for sampling through the inlet on POPS$_{TBS}$.

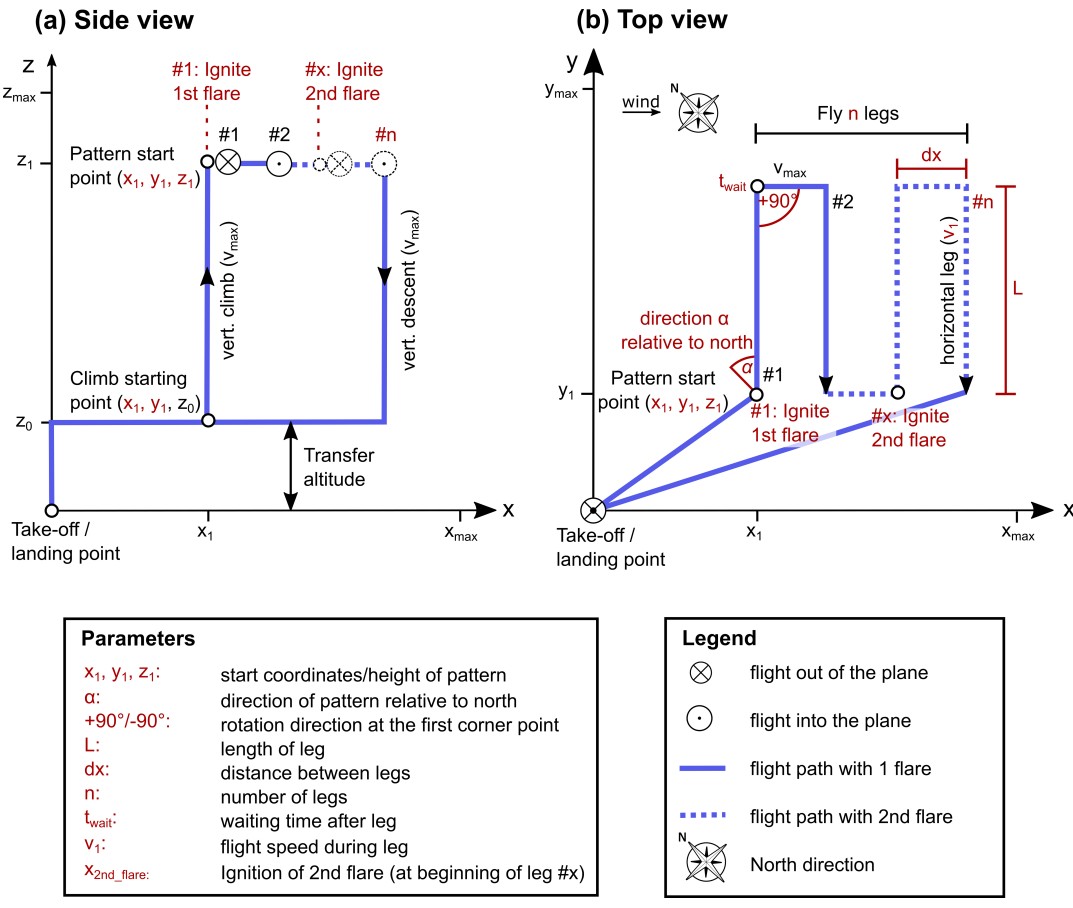

**Figure 3.** A schematic illustrating the parameter space for programming a horizontal seeding mission, in which the seeding UAV flies legs perpendicular to the wind direction (northwest winds here) at the same altitude while seeding, with **(a)** the side view and **(b)** the top view. The same parameter space is used to program vertical seeding missions, in which the seeding UAV flies legs vertically.

## 3 Validation of POPS measurements on the measurement UAV

The POPS has been extensively described, characterized, and validated in previous studies (Gao et al., 2016; Mei et al., 2020; Liu et al., 2021; Mei et al., 2022; Kasparoglu et al., 2022; Pilz et al., 2022; Mynard et al., 2023). Here, we briefly discuss the measurement uncertainties of POPS (Sect. 3.1), quantify the effects that the rotors have on the aerosol measurements (Sect. 3.2 and 3.3), and introduce our new method of ensuring good data quality of high-concentration POPS measurements (Sect. 3.4).

### 3.1 Laboratory-based POPS measurement validations

Before mounting our two POPS onto the UAV and TBS, we performed selected tests in the laboratory to ensure that the two POPS instruments count and size particles correctly. Here, we briefly discuss the main findings of the laboratory tests, while

details of the counting and sizing experiments can be found in Appendix B. To assess the quality of the number concentration measurements, ambient air in the laboratory was simultaneously sampled by the two POPS instruments over a 5-hour period. We found that POPS$_{TBS}$ measured a 5% lower mean particle number concentration than POPS$_{UAV}$ (Fig. B1a) and the values varied by 11% (at the 95% confidence interval) in both instruments. Thus, our results agree with those of Pilz et al. (2022), who found an uncertainty of $\pm 10\%$ for total number concentration. In terms of measuring particle size distributions, the two POPS are in good agreement for most size bins with counting differences below 10% (Fig. B1b). Four size bins (bins 8, 9, 11, and 12) show differences in counts up to 31%, with POPS$_{TBS}$ counting lower values than POPS$_{UAV}$. Furthermore, when measuring monodispersed aerosol of diameters 246 and 522 nm, both POPS correctly size the particles, with a difference in particle number concentrations of 8% (Fig. B2).

In addition, the POPS measurements were compared to measurements from a Scanning Mobility Particle Sizer (SMPS) and an Aerodynamic Particle Sizer (APS). Differences in particle number concentration in the relevant sizes were $28\% \pm 4\%$ compared to the SMPS and $-44\% \pm 8\%$ compared to the APS (Fig. B3). Differences in the size distributions were determined by rebinning the SMPS and APS data to match the respective POPS bin widths and then comparing bin concentrations: differences in bin concentrations between POPS and SMPS and between POPS and APS were both within 70%, except for two outlier bins up to 120% (Fig. B3). However, because these three instruments have different measurement principles, comparing them unavoidably brings additional uncertainty and we cannot know the ground truth. Nevertheless, the measurements agree reasonably well, in line with similar studies by Liu et al. (2021) and Gao et al. (2016).

## 3.2 Comparison of POPS measurements with and without rotors

Previous studies have demonstrated the ability to obtain high-quality aerosol measurements from a POPS mounted on a multirotor UAV (Liu et al., 2021; Brus et al., 2021). Characterizing and validating the measurements obtained from a multirotor UAV is important to quantify any effects that the rotors may have on particle measurements. Since the rotors can produce significant downwash and turbulence (e.g., Ventura Diaz and Yoon, 2018; Jin et al., 2023), the flow into the aerosol inlet may be affected (Alvarado et al., 2017). To assess to what extent the POPS measurements are affected while our UAV is flying, we designed two experiments to compare measurements with and without rotors.

In the first experiment, we compared the POPS particle size distributions measured over 5 minutes, once while the measurement UAV was hovering at approximately 3 m above ground and once while the measurement UAV was standing on top of a trailer with rotors off, also at a height of approximately 3 m above ground (Fig. 4a). Note that the measurements were performed successively. In the second experiment (one hour after the first), we compared measurements from POPS$_{UAV}$ to measurements from POPS$_{TBS}$. Both POPS simultaneously sampled air for 5 minutes at approximately 50 m above ground with approximately 20 m horizontal distance between them (Fig. 4b). In this way, we could compare POPS size distributions from the in-flight UAV to a POPS with no turbulent rotors near the inlet.

When comparing the concentration differences in each size bin during the first experiment at 3 m (Fig. 4c), accumulation mode particles (120-855 nm) are on average within 10%, and coarse mode particles (>855 nm) were undercounted on average by 15% (up to 30%) when the UAV was hovering. These small differences suggest limited effects from rotors in this experiment.

During the second experiment at 50 m, the hovering UAV overcounted particles in both size ranges: accumulation mode particles were on average overcounted by 22% (up to 107%) and coarse mode particles were on average overcounted by 39% (up to 44%). These differences partly arise from comparing two different POPS (whereas the previous experiment uses the same POPS in two modes), especially because the bins with the greatest discrepancies (bins 8, 9, 12, 13, and 14) are some of the bins with the largest differences in the laboratory comparison (Sect. 3.1). Nevertheless, the differences between POPS$_{TBS}$ and POPS$_{UAV}$ while hovering (up to 100%) were larger than the differences between POPS$_{TBS}$ and POPS$_{UAV}$ measured during the laboratory experiments (up to 30 %) (Sect. 3.1). This is most likely due to effects from the UAV rotors. Therefore, we add additional uncertainties of $\pm 22\%$ for accumulation mode particles and $\pm 40\%$ for coarse mode particles for POPS$_{UAV}$ while flying or hovering. However, the differences in mean total particle number concentration were still below 5% for both experiments, indicating that the rotor-induced turbulence has little effect on the total particle number concentration.

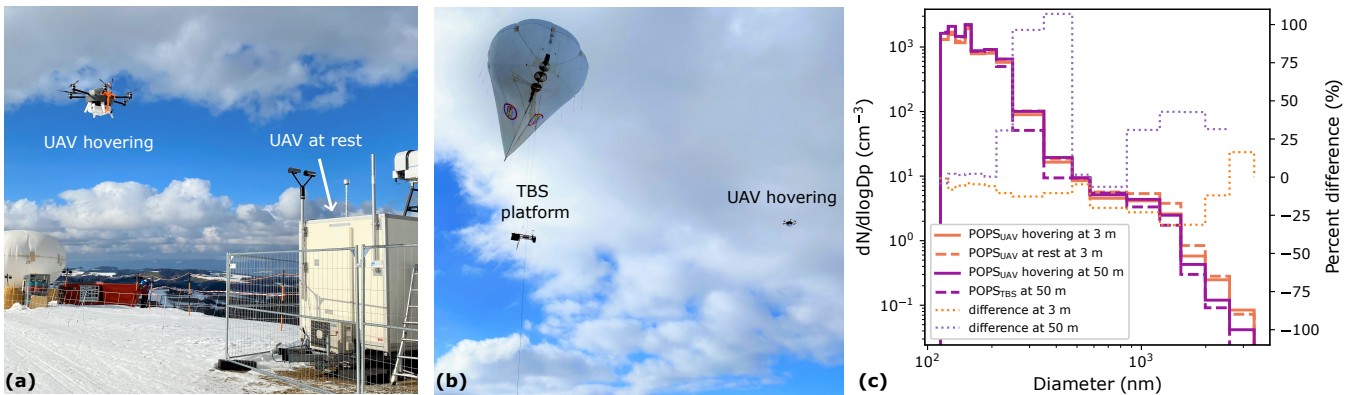

**Figure 4. (a)** Photograph of the rotor comparison experiment at 3 m where the measurement UAV was hovering beside a trailer. The sampling position with static rotors is indicated by the arrow. **(b)** Photograph of the rotor comparison experiment at 50 m where the TBS was flying with its measurement platform containing POPS$_{TBS}$, while the measurement UAV was hovering at a horizontal distance of 20 m from the TBS at the same altitude. **(c)** POPS$_{UAV}$ size distributions from the UAV hovering at 50 m above ground (purple, solid line) compared to size distributions from POPS$_{TBS}$ also at 50 m above ground (purple, dashed line); POPS$_{UAV}$ measurements from when the UAV hovered at 3 m above ground (orange, solid line) compared to when the UAV was 3 m above ground at rest atop the trailer roof (orange, dashed line). Size distributions are measured over the 5-min length of each experiment. The percent difference between each size bin for the experiment at 3 m height (orange dotted line, calculated as ((hovering-rest)/rest $\times$ 100%)) and for the experiment at 50 m height (purple dotted line, calculated as ((UAV-TBS)/TBS $\times$ 100%)) are also shown, corresponding to the right y-axis. The percent difference at 50 m in the largest size bin is not shown (it is undefined) because POPS$_{TBS}$ measures zero counts.

### 3.3 Comparison of POPS measurements during ascending and descending profiles

Another way to identify possible influences of the turbulent downwash from the rotors on the aerosol flow is to compare the ascent and descent measurements of vertical profiles (Liu et al., 2021; Fuertes et al., 2019), because the UAV flies through its own downwash during descent. 34 vertical profiles to 1000 m above ground were performed by the measurement UAV

during the first two winter campaigns. The flights were performed to measure temperature, humidity, wind, and aerosol to plan our seeding experiments (see Section 5), but the flight data can also be used to assess the effect of the turbulent downwash on particle sampling. Particle number concentration measurements were compared between the ascent and descent of each vertical profile (all profiles are in Appendix C). The ascending and descending speed is approximately $10\,\mathrm{m\,s^{-1}}$, thus the total flight time in these profiles is approximately 3 min, and therefore we assume the atmospheric structure to be the same in both directions for any given profile. Qualitatively, the ascent and descent measurements usually agree well with each other under many different atmospheric conditions (Fig. C1). Often the descent flight measurements have more variability than the ascent flight (e.g., in Fig. C1n), likely due to influences of rotor turbulence or flight instabilities in the descent. However, as can be seen in the quantitative assessment described below, this turbulence does not significantly affect the mean concentrations, even over small averaging intervals.

To quantitatively assess differences in the particle number concentration measured during the ascent and descent (Fig. 5), the particle concentration measurements were first binned into altitude intervals of 20 m and then averaged over each interval on the ascent and the descent of each flight. Particle counts from the smallest size bin (115 - 125 nm) were excluded, as it is known that the first size bin has considerably higher inaccuracies (e.g., Mei et al., 2020; Pilz et al., 2022), a common issue with optical particle counters. Additionally, there were 9 outlier data points (out of 9113 total) excluded from the analysis due to extremely unrealistic concentrations (3000 - 50000 $\mathrm{cm^{-3}}$; outliers can be seen in profiles in Figure C1a, l, q, ag).

The mean particle number concentrations of the ascent and descent are in very good agreement across all concentrations and all altitude bins (Fig. 5). The outliers with high descent and low ascent concentrations are the result of a single profile flight with unusual concentrations (see profile in Fig. C1b). Nevertheless, the linear regression of all data has a slope of near unity (0.97) with a Pearson correlation coefficient of 0.90 (p-value < 0.0001). Since the measurements obtained from the ascents and descents are in good agreement, the impact of the rotor downwash on the POPS measurements is negligible when considering particle number concentration.

Because the total particle number concentration is dominated by the high number of accumulation mode particles in comparison to coarse mode particles, we also compared the particle number concentration measured during the ascent and descent considering only the coarse mode particles (Fig. C2). We would expect the measurements of coarse mode particles to be more affected by the UAV rotors compared to the accumulation mode particles based on our previous rotor experiment since small particles generally follow the streamlines of the airflow, whereas large particles have more inertia and can deviate from the streamlines. Therefore, we might expect an enhancement of coarse mode particles in the ascent and a depletion in the descent because the inlet is pointed upwards. However, the number concentration of coarse mode particle concentrations in the ascents and descents are very similar, with the exception of four profiles where the ascents do have higher particle counts (Fig. C2). A quantitative assessment is limited by the fact that there are so few coarse mode particles measured: in nearly all profiles, coarse mode particle counts are below $10\,\mathrm{particles\,s^{-1}}$. The low number results from the generally low number concentrations of coarse mode particles in the atmosphere which may be further reduced by the limited sampling efficiency of supermicron particles during flight in either direction (discussed in Appendix A).

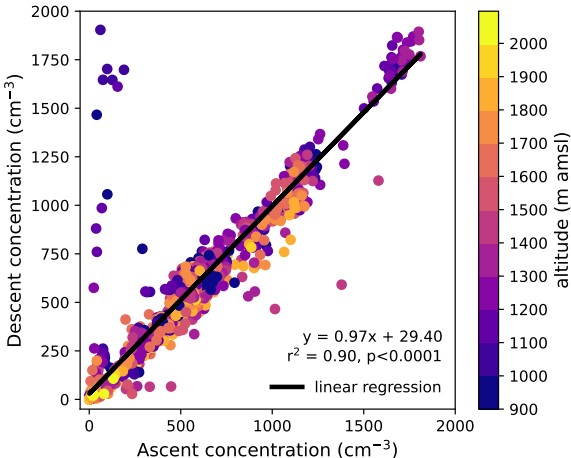

**Figure 5.** Particle number concentrations measured on the ascent versus the descent of 34 vertical profiles by the measurement UAV, colored by their altitude. Measurements from each vertical profile were binned into 20 m altitude intervals, and concentrations were averaged over each altitude interval. The black line is the linear regression through the data.

## 3.4 Data quality filter for POPS measurements at high concentrations

Like every particle counter, POPS has an upper concentration limit above which it does not count and/or size the particles accurately, due to counting limits and coincidence errors (Gao et al., 2016). For POPS, the manufacturer-given range is up to 1000 particles $cm^3$ for less than 10% error, when using a flow rate of $1.67\,cm^3\,s^{-1}$ (Handix Scientific, 2023). However, this concentration is lower than many ambient sampling conditions and is flow rate dependent. Furthermore, there is not yet a common consensus on what happens to the counting accuracy above this concentration range, and which ranges are appropriate for other flow rates. For our purpose, where we use POPS to measure a highly concentrated plume of aerosol (up to 15,000 particles $cm^{-3}$; Sect. 5.1 and 5.2), we developed a new method to filter out "bad" data using the POPS "baseline", a measure of the background scattering signal. Here we detail this data flagging and filtering method which we apply to all POPS measurements to ensure good data quality.

The main source of uncertainty of POPS measurements arises from coincidence errors in particle counting, whereby the scattering signal from one particle overlaps with the scattering signal from the next particle, making it difficult to separate peaks and count two discrete particles. The upper counting limit (software speed limit) of POPS to count every single particle arriving is 10,000 particles $s^{-1}$ (Gao et al., 2016). When using the recommended flow rate of $3\,cm^3\,s^{-1}$, this counting limit corresponds to 3,333 particles $cm^{-3}$ for up to 90% accuracy. Using a lower flow rate of $0.9\,cm^3\,s^{-1}$ (towards the lower end of the possible flow range) results in a counting limit of up to 11,111 particles $cm^{-3}$, although inaccuracies in counting have not been quantified for flow rates other than the nominal recommended flow rate of $3\,cm^3\,s^{-1}$.

In our seeding experiments (described in Section 5), we utilize POPS for measuring the seeding plume. However, the seeding plume is emitted at such high concentrations, that it is difficult to measure it accurately using the standard settings of POPS. In a first set of experiments in early 2022, we flew POPS into the very highly concentrated part of the plume (more than $15{,}000\,\mathrm{cm^{-3}}$), which was approaching/exceeding the upper concentration limit of POPS, highlighting the need for robust filtering to ensure good data quality. In later experiments, we measured the plume further downwind where the plume was more dispersed and concentrations were therefore lower (less than $10{,}000\,\mathrm{cm^{-3}}$). An example timeseries of the particle concentration from an out-of-cloud seeding mission on 9 March 2022 is shown in Fig. 6c, where the measurement UAV was flying horizontal transects through the seeding plume. The particle number concentration was $1{,}500\,\mathrm{cm^{-3}}$ in the atmospheric background and increased up to $14{,}000\,\mathrm{cm^{-3}}$ when the measurement UAV crossed the seeding plume, sampling with a flow rate of $0.9\,\mathrm{cm^3\,s^{-1}}$. The size distributions measurements are shown in Figure 6a, where each timestep and size bin is colored by the bin counts. At many of the timesteps with high concentration measurements, no particles were counted in the smallest size bins ($\approx$ bin 5 and lower). These "holes" in the size distribution heatmap) indicate that particles were not being counted and sized correctly at very high concentrations within the seeding plume. The likely explanation is that, at very high concentrations, a huge amount of coincidence errors occur: there are a high number of large and small particles, and because the large particles have a significantly larger scattering amplitude (scattering amplitude scales with the square of the radius), they block the scattering signals from the smaller particles, thus mainly affect how the smallest particles are sized and counted. It is also possible that many small particles could be miscounted as one larger particle, further reducing the counts in the small-size bins, and also falsely increasing the counts in the large size bins. The missing counts in the small size bins indicate that the true total concentration is likely higher than the $14{,}000\,\mathrm{particles\,cm^{-3}}$ recorded. However, during some timesteps with high total concentrations, there were no "holes" in the size distribution (e.g., at 11:17 - 11:18), suggesting that the true concentration was lower during these timesteps and particles may have been accurately counted and sized. The challenge is then to find which measurements are accurate and which are not.

A parameter that we found useful for assessing the quality of the data is the POPS "baseline" (included in the standard POPS data files) shown in Figure 6b. The baseline is the background scattering signal received by the detector (i.e., a measure of noise in the data) and reported in units of raw analog-to-digital (A/D) counts. A particle is only counted as a particle if its scattering signal is a certain amount larger than the baseline (the default threshold is set to the baseline plus 3 times the baseline standard deviation, e.g., if the baseline is $2000 \pm 5$, then a particle must have a signal of at least 2015 to be counted). When measuring ambient air, the baseline may fluctuate up to $\pm 10$ raw A/D counts from the average. While measuring the seeding plume, however, the baseline can increase up to 800 raw A/D counts higher than the background (Fig. 6b). These increases in baseline correlate with the times when there are "holes" in the size distributions (Fig. 6a): if the (true) total concentration increases, then the baseline will also increase, and at some point, the baseline will be higher than the scattering signal produced by a small particle, such that the small particle will be not be counted. Therefore, we developed a new method for controlling the quality of the POPS measurements using the baseline values: For each seeding experiment, the median of the baseline for the background measurements (not in the seeding plume) was calculated. The threshold for "good data" is set at the background median baseline $+15$, such that any measurement with a higher baseline is flagged and excluded from further

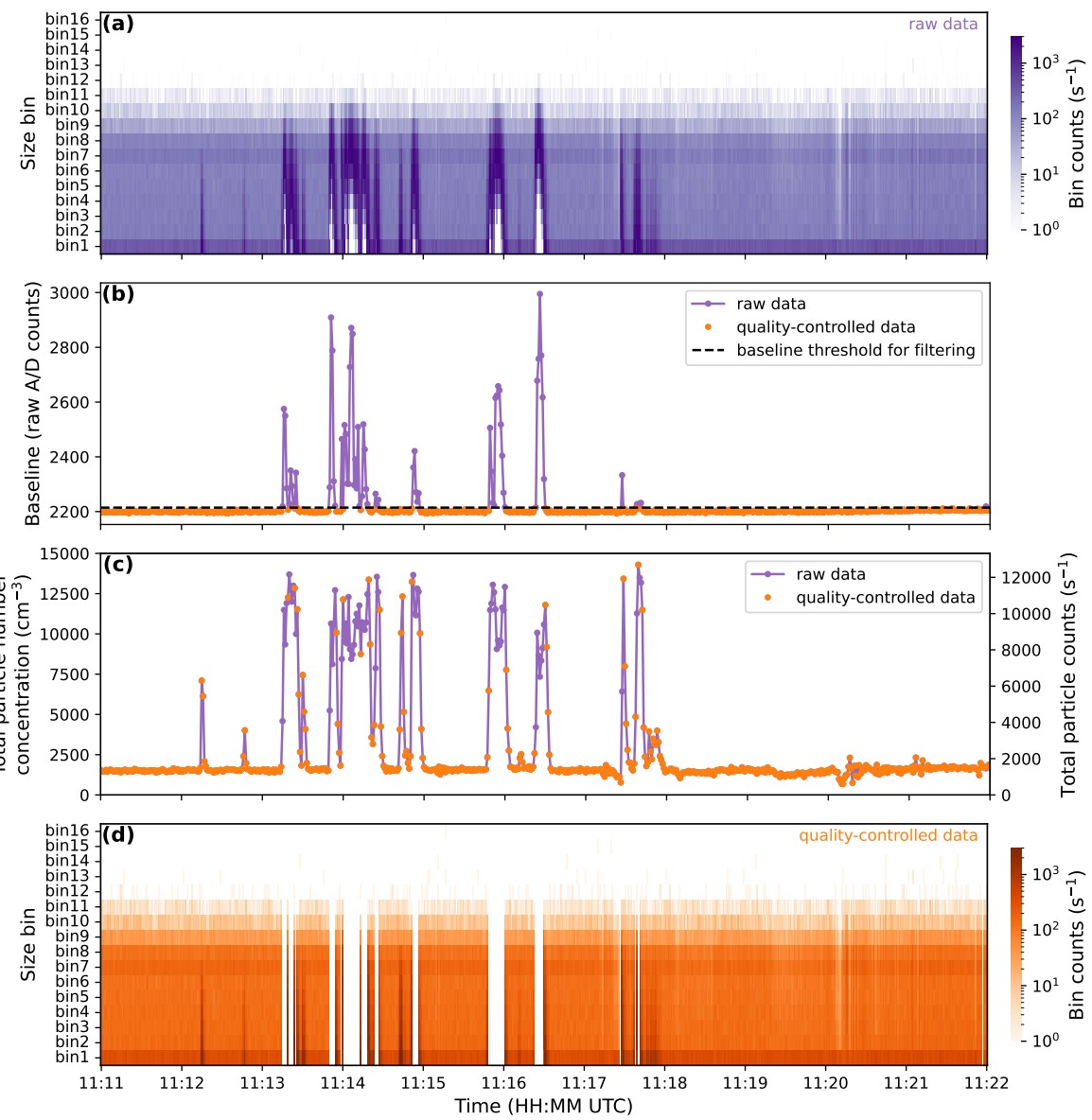

**Figure 6.** Data filtering for an out-of-cloud seeding mission on 9 March 2022. **(a)** Heatmap timeseries of particle number size distributions, with raw data of bin counts per second (purple color scale) in each particle size bin (y-axis) at each 1-second timestep (x-axis). **(b)** Timeseries of the POPS baseline, with raw data (purple) and quality-controlled data after filtering (orange). The black dashed line indicates the baseline threshold value (here, 2214.5 A/D counts) which is applied for filtering the data, i.e., data is excluded for any time when the baseline is higher than the baseline limit. **(c)** Timeseries of total particle number concentration of raw data (purple) and quality-controlled data after filtering (orange). **(d)** Heatmap timeseries of particle number size distributions (like in a), with bin counts per second (orange color scale), for quality-controlled data after filtering.

analysis. In Figure 6b and c, the raw data and the quality-controlled data are both shown to indicate what data passes the filtering. In Figure 6d, an analogous heatmap as in Figure 6a displays the quality-controlled data (after the filter). Many of the high-concentration measurements from within the seeding plume are removed with this approach. We deem the remaining data with high concentration to be trustworthy because the baseline value is within the appropriate range and the size distribution looks reasonable. The case presented here is one of the more extreme cases, and many of our seeding experiments did not require such extensive data removal.

Other studies have suggested applying an upper total concentration limit to filter out bad data. Mei et al. (2022) flagged data with concentrations above $4000\,\mathrm{particles\,cm^{-3}}$, while Mynard et al. (2023) flagged data with concentrations above $7000\,\mathrm{particles\,cm^{-3}}$. However, the concentration measurement itself is biased. The total concentration depends on the counts in each size bin, and if some sizes are categorically not counted, then the concentration will not be reflective of the true concentration. We propose to use the baseline for quality control because it gives a direct indication of whether the background is too high to accurately count particles from all size bins. With this analysis, we also stress the importance of looking into the size distributions for all measurements, and to not only consider total particle concentration.

## 4 Estimating the boundary layer height with the measurement UAV: a case study

One possible application of the measurement UAV is to profile and characterize the planetary boundary layer (PBL), which is of importance for weather predictions and air pollution modeling. There are several methods for determining the height of the PBL, such as by using a vertical profile of relative humidity (RH) or aerosol concentration (e.g., Summa et al., 2022; Jozef et al., 2022), both of which can be obtained from a measurement UAV profile (Hervo et al., 2023). We present one example of a vertical profile up to $1000\,\mathrm{m}$ above ground by the measurement UAV (flight speed of $10\,\mathrm{m\,s^{-1}}$) on 8 March 2022 at 14:28 UTC at the CLOUDLAB main site. The mean RH profile (Fig. 7a) from ascent and descent shows a sharp decrease in humidity between 1350 and $1550\,\mathrm{m}$ amsl, where RH decreases from 60% to 20%. The height of the PBL using this RH profile can be estimated by finding the minimum (i.e., most negative) RH gradient with respect to altitude (Seidel et al., 2010; Collaud Coen et al., 2014), which results in a PBL height of $1421\,\mathrm{m}$ amsl.

Similarly, we can also use the particle number concentration profile from $\mathrm{POPS_{UAV}}$ to determine the PBL height by again finding the minimum in the gradient of concentration with respect to altitude. The PBL was calculated at $1467\,\mathrm{m}$ amsl for the ascent flight and at $1426\,\mathrm{m}$ amsl for the descent flight (Fig. 7b). The PBL heights derived from the POPS measurements are associated with an uncertainty of $\pm 20\,\mathrm{m}$ because 1) the sampling frequency of POPS ($1\,\mathrm{s^{-1}}$) multiplied by the flight speed ($10\,\mathrm{m\,s^{-1}}$) gives a sampling resolution of $10\,\mathrm{m}$, and 2) the GPS altitude measurements have an estimated uncertainty of $10\,\mathrm{m}$. Therefore, the POPS-derived PBL heights from the ascent and descent are in good agreement with each other and with the RH-derived PBL height.

To further validate these PBL height estimates, we compare the profiles to simultaneous co-located ceilometer measurements of the attenuated backscatter (Fig. 7c). Qualitatively, the particle number concentration profiles measured by the UAV are similar to the profile measured by the ceilometer: both measurements indicate high aerosol concentrations below $1400\,\mathrm{m}$ amsl

and a sharp decrease above it, as well as a thin layer of elevated aerosol concentrations at approximately 1650 m amsl. The
PBL height calculated from the ceilometer data using the manufacturer's algorithm (Lufft) was at 1440 m amsl during both
the ascent and descent of the UAV flight. The ceilometer-derived PBL height is in good agreement with the RH-derived and
POPS-derived PBL heights (±20 m) and the differences are very small compared to the general disagreement between PBL
detection methods (Collaud Coen et al., 2014). This case study illustrates that the measurement UAV can characterize the lower
atmosphere similarly to a ceilometer, with the advantage that it has co-located meteorological measurements.

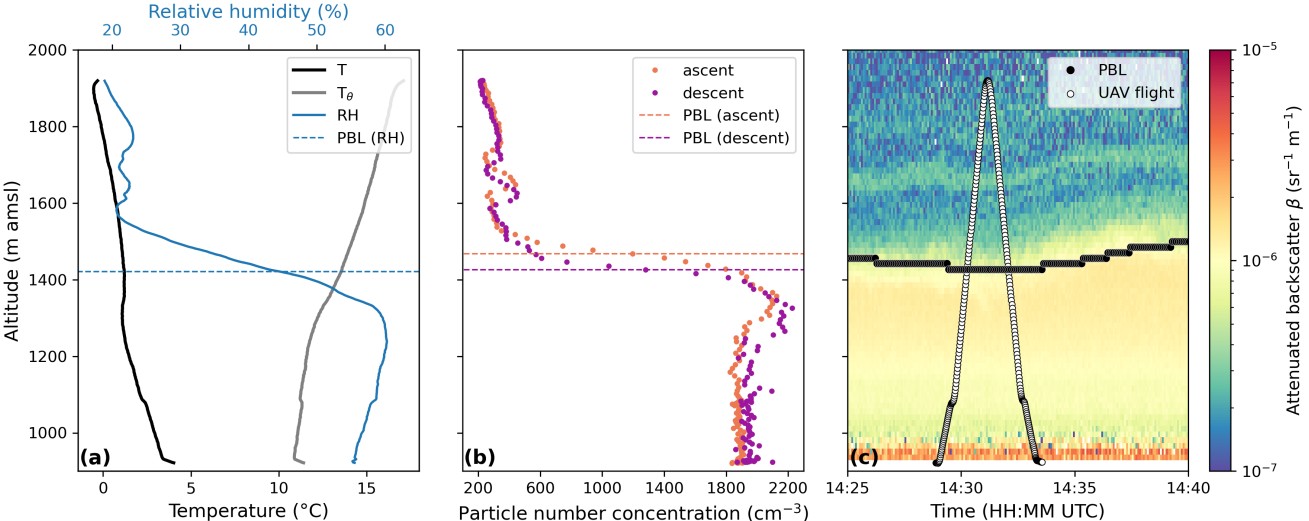

**Figure 7.** Height-resolved meteorological and aerosol properties observed by the measurement UAV on 8 March 2022 at 14:28 UTC, com-
pared to co-located ceilometer backscatter data. **(a)** Temperature (black), relative humidity (blue), and potential temperature (grey) profiles
with the measurement UAV. The horizontal dashed blue line at 1421 m indicates the PBL height derived from the RH gradient. **(b)** POPS$_{UAV}$
particle number concentrations measured during the ascent (orange dots) and descent (purple dots) flight. Horizontal dashed orange and
purple lines at 1467 m and 1426 m indicate the PBL heights derived from the gradient of the ascent and descent particle concentration,
respectively. **(c)** Attenuated backscatter $\beta$ time series measured by the ceilometer, with black circles indicating the detected PBL height
obtained from the manufacturer's algorithm and white circles indicating the UAV flight path.

## 5   Application of UAVs for seeding experiments

We have shown that multi-rotor UAVs can be used for injecting seeding particles in the atmosphere and for accurately and
flexibly measuring aerosol in the lower atmosphere. Next, we demonstrate how the seeding and measurement UAV are de-
ployed within the CLOUDLAB project (Henneberger et al., 2023) by presenting selected examples. First, we show how the
measurement UAV with POPS$_{UAV}$ can be used to characterize the dispersion of an out-of-cloud seeding plume (Section 5.1 and
Fig. 8a). The purpose of the out-of-cloud seeding experiment was to estimate the concentration and dispersion of the particles

produced from the flares onboard the seeding UAV. Second, we present an in-cloud seeding experiment in a supercooled stratus cloud where the changes in the aerosol and microphysical properties induced by the seeding UAV were measured downstream by the TBS (Section 5.2 and Fig. 8b). The in-cloud seeding experiment was designed to induce ice nucleation and observe ice crystal growth in supercooled clouds. The examples presented here demonstrate the capabilities of the UAVs and other instrumentation, and further results will come in future publications.

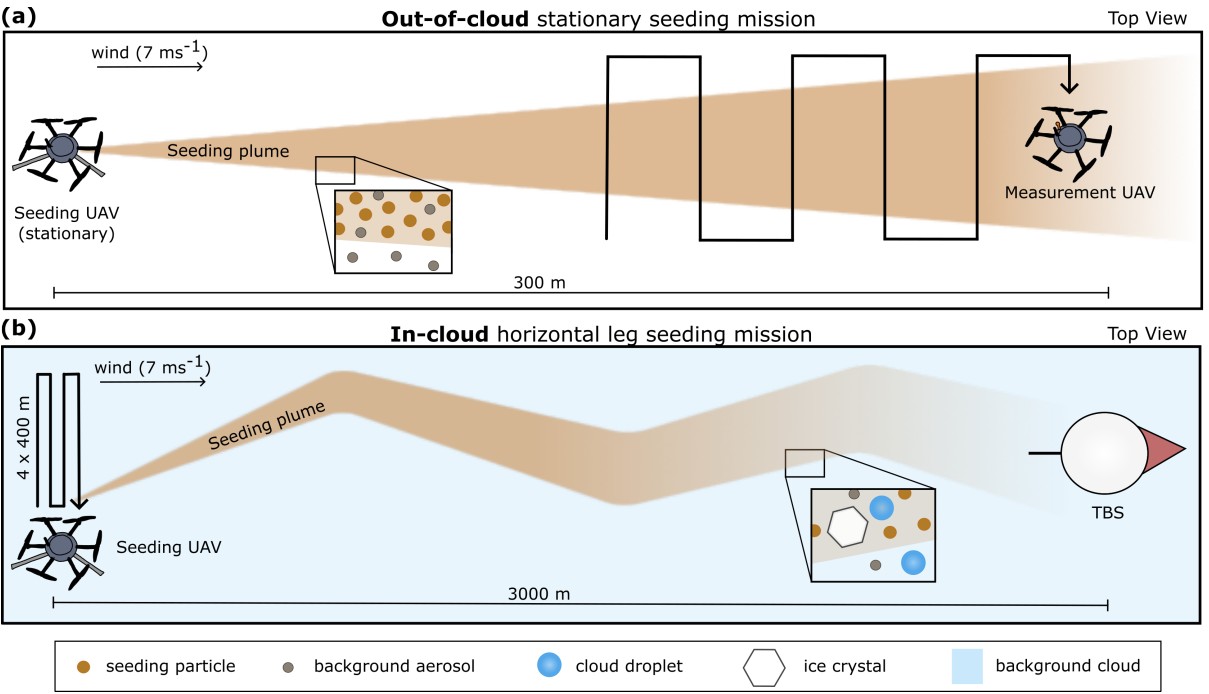

**Figure 8.** Illustration of two example seeding missions from a top-down view (not to scale). **(a)** An out-of-cloud stationary seeding mission, in which the seeding UAV hovers stationary at a constant altitude while burning a flare, while the measurement UAV flies horizontal transects through the plume. The inset illustrates that the seeding plume contains seeding particles and background aerosol, whereas outside the plume there is only background aerosol. **(b)** An in-cloud horizontal leg seeding mission (blue background is the background cloud), in which the seeding UAV flies 4 horizontal legs of each 400 m, all 3000 m upstream of the TBS. The distance between legs is shown for illustration purposes; often the legs are performed at the same location. The inset illustrates that the seeding plume contains seeding particles, cloud droplets, ice crystals, and background aerosol, whereas outside the plume there are background aerosol and cloud droplets. In all experiments, the UAVs and the TBS fly at the same altitude.

## 5.1   Characterizing an out-of-cloud seeding plume with POPS on the measurement UAV

During an out-of-cloud stationary seeding mission (illustrated in Fig. 8a), the seeding UAV burns 1-2 seeding flares while hovering stationary at the defined altitude, while the measurement UAV flies horizontal legs through the seeding plume. These missions can be flown autonomously, but here we present a case in which the measurement UAV was manually controlled

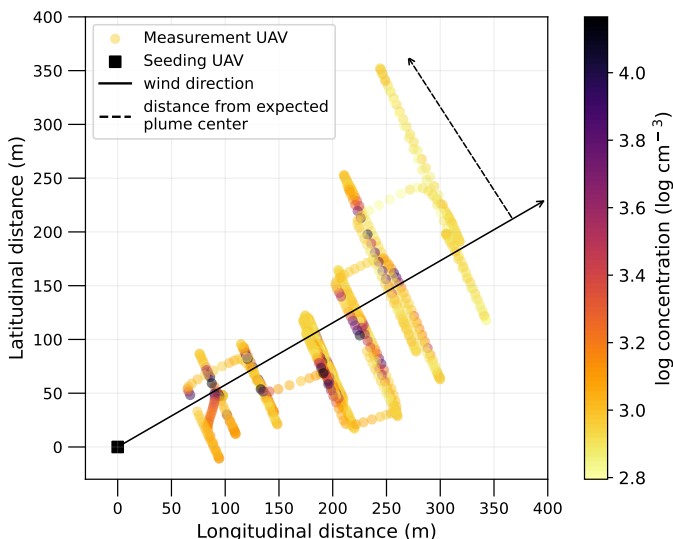

**Figure 9.** The flight path of the measurement UAV in relation to its longitudinal and latitudinal distance from the seeding UAV (black square), colored by the particle number concentration measured by POPS$_{\text{UAV}}$, during the out-of-cloud seeding mission on 28 March 2022. The solid line arrow indicates the mean wind direction during the mission. The dashed arrow shows distance from expected plume center, which is used as the x-axis in Figure 10.

by an experienced and properly educated pilot (Fig. 9). This seeding mission was performed on 28 March 2022 at 9:30 UTC under clear-sky conditions. Seeding altitude was 1320 m amsl, with a temperature of 9.5 °C, a wind speed of 7 m s$^{-1}$, and a wind direction of 240° (measured by radiosonde and UAV profile). The seeding UAV hovered at the seeding altitude while two consecutive flares burned, while the measurement UAV flew transects perpendicular to the wind direction at six different

distances (80 - 370 m) downwind of the seeding location (Fig. 9). At each distance downwind, between two and nine legs were flown through the plume. Because of the small distance between the seeding location and the measurement UAV, the plume was highly concentrated (more than 15,000 particles cm$^{-3}$) and thus required data filtering (31 of 104 in-plume data points were removed), according to the method introduced in Section 3.4.

The concentration-colored flight path (Fig. 9) shows that the concentrations measured inside the plume (downwind of the
400 seeding UAV along the main wind direction) exceeded the background concentrations (1000 cm$^{-3}$) by up to 1.5 orders of magnitude. However, there was significant variability in the location and magnitude of concentration peaks. This variability in the plume measurements becomes apparent when viewing the concentration as a function of distance from the expected plume center line (see arrow in Fig. 9) for four downwind distances (144, 210, 250, and 300 m) (Fig. 10). First, note that not all transects measured concentrations above the background, indicating that some of the transects were not actually passing
through the plume. Therefore, the plume itself must have been displaced horizontally and/or vertically because the transects at each downwind distance were flown at the same location. Plume displacement is also evident when considering the center

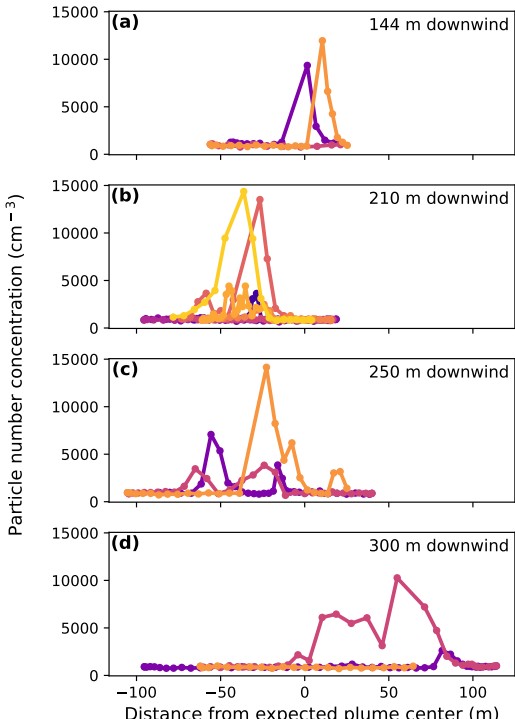

**Figure 10.** Particle concentrations measured inside the seeding plume as a function of distance from expected plume center (see arrows in Fig. 9), for the out-of-cloud seeding mission on 28 March 2022. Concentration was measured by POPS$_{\text{UAV}}$ while the measurement UAV flew horizontal transects perpendicular to the wind direction through the seeding plume at several distances downwind of the seeding UAV: (a) 144 m, (b) 210 m, (c) 250 m, (d) 300 m. For 210 m distance nine transects were made, for the other distance only three. The colors are used for distinguishing the individual transects.

of the plume peaks – most are not centered at 0 m. At 300 m downwind, the peaks are horizontally displaced by 20 or 60 m. Second, there are considerable dissimilarities in the width and height of each concentration profile. These dissimilarities oc-cur both between transects at the same downwind location and between different downwind locations. Interestingly, there is 410 no consistent trend with increasing downwind distance in terms of concentration magnitude or peak width, contrary to what would be expected according to Gaussian dispersion (i.e., decreasing concentration and increasing peak width with increasing distance downwind). These measurements indicate the turbulence within the seeding plume and generally illustrate the unpre-dictable nature of the dispersion of particles in a plume. Indeed, because of the complexities of turbulence, accurately modeling atmospheric particle dispersion is known to be difficult (e.g., Shirolkar et al., 1996; Holmes and Morawska, 2006), especially 415 on small spatial and time scales as we measure here. The method presented here provides a potential framework for further quantitative experimental investigations into aerosol dispersion, relevant for air pollution modeling and other applications.

## 5.2 Characterizing an in-cloud seeding plume with POPS mounted on the TBS

In an in-cloud horizontal leg seeding mission (Fig. 8b), the seeding UAV flies horizontal legs perpendicular to the wind direction upstream of the TBS within a supercooled cloud. Because the seeding pattern is perpendicular to the wind direction, the seeding plume creates a zig-zag shape as it gets advected toward the TBS, and the signal measured by the TBS is then expected to be multiple distinct signals corresponding to each of the seeding legs. The seeding plume in-cloud is expected to contain a mixture of supercooled cloud droplets (from the pre-existing cloud and/or newly created droplets from seeding particles that activated as cloud condensation nuclei), ice crystals (from the pre-existing cloud and/or nucleated by seeding particles acting as ice nucleating particles), and the remaining un-activated/un-nucleated seeding particles. $POPS_{TBS}$ measures these leftover seeding material particles, while the holographic imager HOLIMO measures the cloud droplets and ice crystals. Although the measurement UAV was not operated during in-cloud seeding missions in the CLOUDLAB campaigns of 2021/22 and 2022/23 due to logistical reasons, it can be used as an additional measurement platform in future campaigns to characterize the in-cloud seeding plume in between the seeding UAV and the TBS.

The in-cloud seeding mission we present here was conducted on 24 January 2023 at 19:45 UTC. At that time, the measurement site was covered with a persistent stratus cloud with a cloud base at approximately 1000 m amsl (measured by the ceilometer) and a cloud top at 1600 m amsl (measured by the cloud radar). The seeding altitude was chosen as 1350 m amsl, with a temperature of $-5.1\,^\circ$C (measured by the seeding UAV). At the seeding height, the wind direction was $77^\circ$ and the wind speed was $7\,\mathrm{m\,s^{-1}}$ (measured by the radar wind profiler and a radiosonde). The seeding UAV flew four 400 m legs, with no distance between legs, 3000 m upwind of the TBS measurement platform (similar to Fig. 8b). The seeding flare ignited at 19:44:46 UTC and the seeding pattern ended at 19:50:26, for a total estimated burning time of 5 minutes and 40 seconds.

After this seeding mission, the TBS was brought back to ground very soon after the experiment (see altitude of TBS, Fig. 11b), thus allowing measurements of three different environmental conditions in a short period of time: the background supercooled stratus cloud, the seeding plume in-cloud, and the background below the cloud. In the particle number concentration measurements (Fig. 11a), the seeding plume signal (370 - 800 $\mathrm{cm^{-3}}$) stands out clearly from the in-cloud background (100 - 340 $\mathrm{cm^{-3}}$) during the passage of the seeding plume. The seeding signal is also visible in the ice crystal number concentrations, which increase from 0 up to 500 $\mathrm{L^{-1}}$ (Fig. 11b) at the same time as the particle number concentration increases. Four distinct groups of peaks can be seen in the concentrations, corresponding to the four legs of the seeding pattern. The first signal appears at 19:52:39 and the last signal ends at 19:56:22, for a total duration of 3 minutes 43 seconds, starting 7 minutes 53 seconds after the flare was ignited. Based on the estimated local wind speed and the distance between seeding and measuring, the calculated advection time of the seeding particles is 6 minutes 58 seconds; i.e., we would expect to see the signal in $POPS_{TBS}$ approximately 7 minutes after seeding started, in the absence of turbulence. Therefore, we assume the elevated concentrations that $POPS_{TBS}$ and HOLIMO measured are the seeding plume passing by, and not natural variation in the cloud. Furthermore, small deviations in the timing compared to the calculated timing are expected due to uncertainties in wind measurements as well as variability and turbulence in the 3000 meters between seeding and measuring. Turbulence and mixing within the cloud are also demonstrated by the fact that there is significant variability in the particle and ice crystal number concentrations and

the time spans of the seeding signals (Fig. 11), similar to the findings from the out-of-cloud seeding case discussed previously (Section 5.1).

Particle size distributions for each of the three situations (plume in-cloud, background in-cloud, and background below-cloud) are shown in Figure 11c. The seeding plume had 2 - 10 times more particles with sizes between 165 and 1220 nm compared to the in-cloud background, but a similar number of particles of size $> 1220$ nm. In contrast, the below-cloud background had 6 - 75 times fewer particles $> 1220$ nm than the in-cloud, but 2 - 60 times more particles $< 1220$ nm. These size distributions indicate that the $> 1220$ nm particles POPS$_{TBS}$ measured in-cloud were likely small cloud droplets, since they were not present in the below-cloud measurement and were present in similar amounts in both the in-cloud seeding plume and in-cloud background. It is also notable how the total particle number concentration in the below-cloud measurement (approx. 700 cm$^{-3}$) was significantly higher than the in-cloud background (up to 340 cm$^{-3}$) showing the effects of particle activation into cloud droplets as well as scavenging of aerosol particles by cloud droplets, as previously documented by others (e.g. Flossmann and Wobrock, 2010; Ohata et al., 2016).

Finally, it is important to note that the in-cloud background particle concentrations had large fluctuations in concentration (50 - 340 cm$^{-3}$, Fig. 11a). These fluctuations were present in POPS$_{TBS}$ in-cloud measurements in around half of all the in-cloud seeding missions and were likely caused by moisture build-up in the POPS inlet. The moisture may interfere with the air inflow or with the optical measurement itself. The issue can be solved by running POPS in clean, dry conditions for a few minutes between experiments. When taking these additional measures in our next campaign, we can obtain more consistent measurements. Nonetheless, in the measurements we have so far, this issue was usually not severe enough to mask the seeding signal from the background. For future projects, it could be worthwhile to build an inline drying or heating mechanism in the inlet, with the consequent exclusion of cloud droplet measurements due to their evaporation.

## 6 Discussion and Conclusions

This paper presented two new UAVs: a seeding UAV equipped with burn-in-place flares and a measurement UAV equipped with a Portable Optical Particle Spectrometer, both able to fly into supercooled clouds. We introduced the flight patterns of the measurement and seeding UAV with the parameter space available to configure the flight missions (Sect. 2.4). We then showed that the POPS data are comparable to other aerosol instrument measurements (particle number concentrations within 50%; Sect. 3.1) and that there is a minimal effect of rotor-induced turbulence from the UAV on particle number concentration (Sect. 3.2 and 3.3). We also developed a new method for filtering out high-concentration data based on the dynamic baseline of the POPS (Sect. 3.4). Finally, we presented measurements from selected experiments to demonstrate how we can successfully measure the boundary layer (Sect. 4) and a seeding plume in and out-of-cloud (Sect. 5.1 and 5.2). We see the following three major applications, discussed below.

First, the measurement UAV can be used for profiling the atmosphere, i.e., measuring temperature, humidity, wind, and particle number concentrations. In Section 4, a measurement UAV profile was compared to the backscatter measurements from a ceilometer, showing a similar trend in POPS$_{UAV}$ particle concentrations as the ceilometer with respect to height. This case

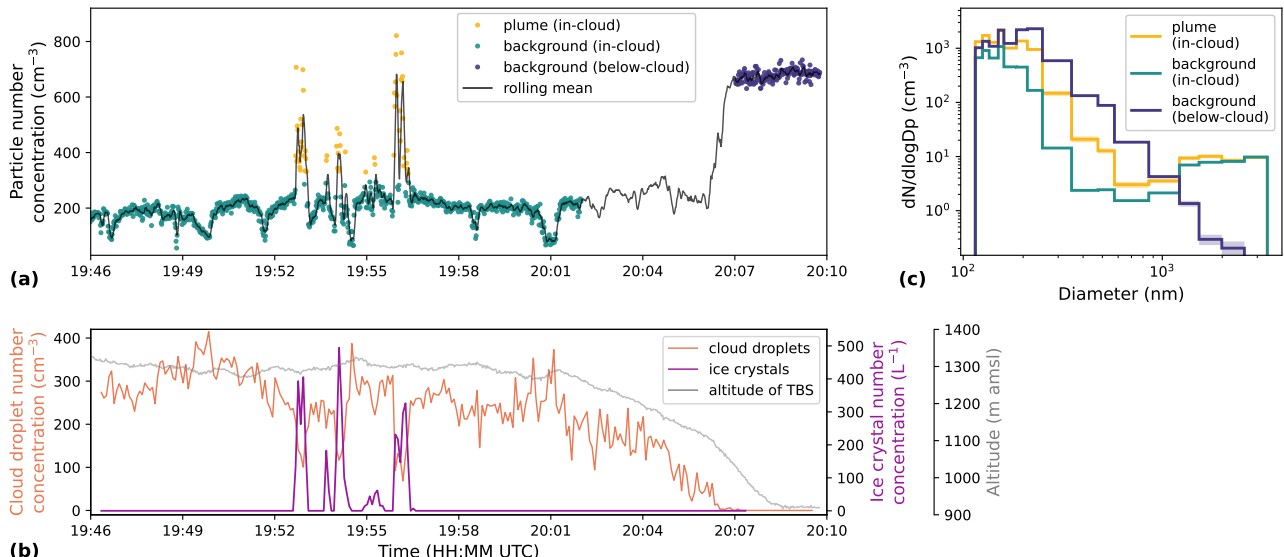

**Figure 11. (a)** Time series of total particle concentration measured by POPS$_{TBS}$ (solid black line is a 5-second rolling mean of the 1-second data points) from the in-cloud horizontal leg seeding mission on 24 January 2023 when the seeding UAV flew 4 legs of each 400 m, 3000 m upwind of the TBS measurement platform. Yellow markers indicate when POPS$_{TBS}$ measured the seeding plume in-cloud (defined as total concentration $> 370 \, \mathrm{cm}^{-3}$ while in cloud), green markers for when POPS$_{TBS}$ was measuring the background in-cloud (total concentration $< 340 \, \mathrm{cm}^{-3}$ while in cloud), purple markers for when POPS$_{TBS}$ measured background below-cloud, and no markers for when the TBS was transitioning between altitudes (for altitude, see (c)). **(b)** Cloud droplet number concentration (orange line, left y-axis) and ice crystal number concentration (magenta line, first right y-axis), as measured by the holographic imager HOLIMO aboard the TBS, are shown for the same period as the aerosol measurements. The grey line and second right y-axis show the corresponding altitude (m amsl) of the TBS during this period. **(c)** Size distributions of the seeding plume in-cloud (yellow), the background in-cloud (green), and the background below-cloud (purple), corresponding to the data markers in panel a. Shading around the mean represents the standard error.

demonstrates how the UAV can serve as a more flexible alternative for characterizing the lower atmosphere. Additionally, the propeller heating and the flight time of around 20 minutes allow for flights up to 6 km amsl including into supercooled clouds. Profiling the atmosphere with in situ measurements is important for understanding and predicting local air quality and health effects, atmospheric transport, and boundary layer meteorology, for which our measurement UAV is a useful tool.

The second application is the characterization of an aerosol plume. Our measurement UAV can fly autonomous measurement missions (Section 5), where it can fly horizontal or vertical transects through a stationary plume, or hover stationary while a plume is passing. In the CLOUDLAB project, we use this approach to characterize the cloud seeding plume, though the UAV can easily be used for characterizing any other type of plume, such as from a factory chimney. The data obtained from such plume dispersion measurements could help to better map, model, and predict the dispersion and transport of pollution in our atmosphere.

Finally, the third, and most novel application is glaciogenic cloud seeding with our seeding UAV. We showed that it can burn a flare containing around 20 g of ice-active seeding material, directly in stratus clouds with ambient temperatures below $-5\,^\circ$C. Because our UAVs have a propeller heating system to prevent ice buildup, they are capable of flying in such supercooled clouds, which has so far been a major challenge for the use of UAVs in cloud research. CLOUDLAB's cloud seeding experiments were primarily designed for the purpose of investigating ice crystal formation and growth (Henneberger et al., 2023), so it is essential for us to have the ability to seed directly within supercooled clouds, where ice nucleation initiates almost immediately. Its feasibility for operational seeding has not been investigated here and is not a goal of CLOUDLAB. Rather, our seeding method is ideal for researching the microphysical processes of aerosol-cloud interactions and ice crystal growth within persistent stratus clouds. We have shown that not only can we produce a cloud seeding plume from a multirotor UAV, but we can also detect seeding particles and ice crystals up to 3000 m downstream (Sect. 5.2), and in future work we can therefore assess the microphysical changes within the plume. We also explained our control over parameters like seeding distance, height, and pattern extent. Future work will include using these methods to quantify ice crystal formation and growth in real cloud conditions, as well as to investigate the aerosol-cloud interactions by these seeding particles, namely their hygroscopic growth, cloud droplet activation, and ice nucleating abilities.

## Appendix A:  Sampling efficiency of POPS inlet

With any aerosol measurements, it is important to consider the particles' sampling efficiency through the inlet and tubing. The sampling efficiency of the system can be estimated by multiplying the aspiration efficiency, which refers to how particles enter the inlet from the ambient air, by the transport efficiency, which refers to how particles are transported through the inlet tubing to the instrument (Brockmann, 2011). The aspiration efficiency depends on the inclination angle of the inlet with respect to the ambient air, as well as the relative velocities of the inlet flow to the ambient air flow. The transport efficiency depends on factors such as gravitational deposition of larger particles, diffusional loss of smaller particles, and the number and angle of bends in the tubing.

It is challenging to comprehensively assess all of the relevant factors and precisely calculate the sampling efficiencies of our system. For POPS$_{UAV}$ in particular, the sampling efficiency depends further on the flight behavior of the UAV and on the ambient conditions. Whether the UAV is hovering, flying horizontally, ascending, or descending, and the speed at which it is flying, as well as the horizontal and vertical wind motions of the air, all directly impact the aspiration efficiency. Therefore, here we apply certain assumptions and simplifications to obtain a base estimate of the sampling efficiencies.

Our inlet on POPS$_{UAV}$ (seen in Figure 1a; the same inlet is on the POPS$_{TBS}$) consists of a 25 cm long brass tube (2 mm inner diameter, 3 mm outer diameter) facing upward on the UAV, with a 90° bend and 3.5 cm long horizontal section that directs into the instrument. On top of the inlet, there is a small cap which is intended to block very large particles, cloud droplets, and ice crystals from directly entering the inlet from above. Therefore, all particles must make two bends around the cap to enter the inlet in order to be sampled. To simplify calculations, we only consider particles of 100 nm and 3 μm diameter, in order to estimate the sampling efficiencies for the lower and upper bound of the POPS size range. We assume a particle density of

$2\,\mathrm{g\,cm^{-3}}$, consistent with ambient air estimates (Thomas and Charvet, 2017), and a constant flow rate of $3\,\mathrm{cm^3\,s^{-1}}$, which gives an inlet flow velocity of $0.95\,\mathrm{m\,s^{-1}}$. All calculations use the equations found in Brockmann (2011).

First, we consider the transport efficiencies of the POPS inlet. For 100 nm particles, the main losses occur due to diffusion
through the full length of the tube (28.5 cm), resulting in approximately 1% loss, or a transport efficiency of 99%. For 3 μm particles, gravitational deposition leads to transport losses in the horizontal section of the tube (approx. 9% loss) and in the 90° bend (approx. 5% loss assuming laminar flow), resulting in a transport efficiency of 86% (i.e., $0.91 \times 0.95 \times 100 = 86\%$). Transport efficiencies will be the same for both POPS$_\mathrm{TBS}$ and POPS$_\mathrm{UAV}$ regardless of the flight behaviour of the UAV/TBS or the ambient conditions.

Next, we consider the aspiration efficiencies of the POPS inlet. For a 100 nm particle, the aspiration efficiency is around 100% (within approx. 1%) independent of the environmental conditions, because small particles follow the streamlines of the airflow due to their little inertia. For 3 μm particles, the inertia is sufficiently large that the particles can diverge from the air streamlines, thus the aspiration efficiency can deviate strongly from 100%, depending on the flow conditions and the sampling angle, as described in the following.

If we consider a simplified case with no horizontal wind and the UAV ascending at $10\,\mathrm{m\,s^{-1}}$, we have an ambient air velocity with respect to the inlet of $10\,\mathrm{m\,s^{-1}}$ (Fig. A1a). Because of the cap on top of the inlet, we assume the particles must make two bends in order to enter the inlet: the first bend to get into the space underneath the cap, and the second bend to enter the inlet tube. We can estimate the transport through the first bend as if it were a bend in a tube: we assume the particles make a 90° bend at the velocity of the air ($10\,\mathrm{m\,s^{-1}}$) in a "tube" with diameter equal to the space between the inlet and the cap (9 mm) (Equation
6-66 in Brockmann, 2011). For a 3 μm particle, this gives a loss of 26% in the first bend. Since the second bend results in the particle entering the inlet, we must estimate that with the aspiration efficiency equation for sampling at a given angle (90°) of the inlet with respect to ambient air, which takes into account the relative velocities of the ambient air ($10\,\mathrm{m\,s^{-1}}$) and the inlet flow ($0.95\,\mathrm{m\,s^{-1}}$) (Equation 6-22 in Brockmann, 2011). Although this equation is not valid for our angle and velocity regime because it is out of the range of the empirical data for which the equation is based on, we can still use this to see that
the aspiration efficiency approaches 0% for similar inlet situations. For a descending UAV with no horizontal wind (Fig. A1b), the calculations for 3 μm particles are analogous to before because again the particles must make two bends to enter the inlet, thus also giving aspiration efficiencies approaching 0%.

If we now consider when the UAV is hovering, with a small vertical air flow velocity of $2\,\mathrm{m\,s^{-1}}$ (from the flow created by the rotors), we would get an aspiration efficiency of 54% for 3 μm particle entering the inlet after both bends. If we consider
a lower inlet flow rate of $0.9\,\mathrm{cm^3\,s^{-1}}$, which we also sometimes used for POPS$_\mathrm{UAV}$, then the aspiration efficiency would again approach 0% in this last considered case. Finally, if we consider non-zero horizontal wind speeds, we assume the initial bend would be larger than 90° because the air comes at an angle relative to the inlet (Fig. A1c), which further increases the loss in that bend, thus further reducing the theoretical aspiration efficiency.

Overall, these simplified calculations indicate that we should not be able to measure supermicron particles while the UAV
is flying. However, our measurements during profiling show that we measure supermicron particles up to $15\,\mathrm{s^{-1}}$ (see Figure C2). The discrepancy likely originates from overly simplified calculations for our system, which serve as a conservative limit.

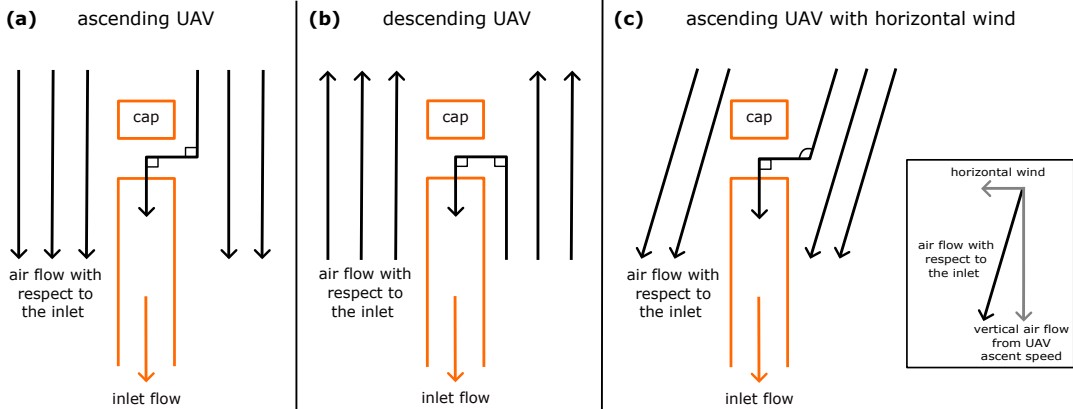

**Figure A1.** Schematic of the POPS$_{UAV}$ inlet and relevant air flows (not to scale), under the simplifications and assumptions used to calculate inlet sampling efficiency. Sampling conditions for **(a)** an ascending UAV, **(b)** a descending UAV, and **(c)** an ascending UAV with horizontal wind.

We hypothesize that one important factor missing in the calculations is the turbulence created by the UAV rotors. Turbulence makes the air flows go in varying directions and speeds, thereby affecting the angles and flow velocities towards the sampling inlet, and likely increasing the likelihood that large particles can be sampled. Because the inlet top is 5 cm above the height of
the rotors, most of the rotor downwash is avoided, but still turbulence and general air flow disturbances can extend a couple of meters above the rotors (Jin et al., 2023). Computational fluid dynamics simulations would be needed for more complete and valid estimates of sampling efficiencies.

## Appendix B:  Laboratory measurement validations of POPS

Three laboratory-based validation experiments are presented here: (1) a comparison of the two POPS instruments measuring
ambient polydisperse air (Fig. B1), (2) measurements of lab-generated monodispersed particles (Fig. B2), and (3) a comparison to reference instruments measuring polydisperse ambient air (Fig. B3). It was not our intention to perform detailed or extensive characterizations of POPS, as these have been reported previously (Gao et al., 2016; Mei et al., 2020; Liu et al., 2021; Pilz et al., 2022). Our goal was to ensure good performance of both POPS instruments in terms of counting and sizing particles.

In the first experiment, POPS$_{UAV}$ and POPS$_{TBS}$ simultaneously measured polydisperse ambient laboratory air over 5 hours.
Differences in particle number concentration at 1 second time resolution reveal that POPS$_{TBS}$ consistently measured slightly lower concentrations than POPS$_{UAV}$ (Fig. B1a). The mean difference in particle number concentration between the two POPS was $5 \pm 11\%$ (at the 95% confidence interval). When comparing size distributions, we see that for nearly all size bins, the differences are within 10% between the two POPS, with four size bins reaching a 31% difference (Fig. B1b).

In the second experiment, size validations for POPS were performed by measuring monodispersed particles of three differ-
ent sizes (246 nm, 522 nm and 3 μm, Figure B2). The submicron particles of 246 nm (Fig. B2a) and 522 nm (Fig. B2b) were

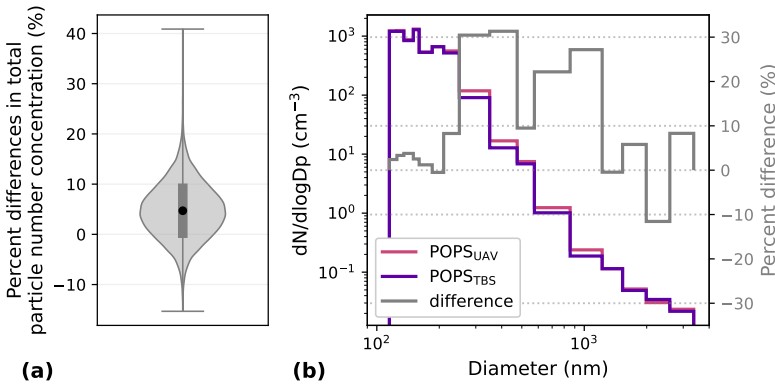

**Figure B1. (a)** Violin plot of the percent differences in total particle number concentration between POPS$_{TBS}$ and POPS$_{UAV}$ measuring ambient lab air for 5 hours, at a 1 second time resolution (sample size is 18,214). Black circle is the mean, and the box edges are at one standard deviation. **(b)** Size distributions of POPS$_{TBS}$ (purple) and POPS$_{UAV}$ (magenta) during the same 5-hour measurement of ambient air. Percent differences (grey, right y-axis) between POPS$_{TBS}$ and POPS$_{UAV}$ were calculated for each bin. Differences are calculated as ((UAV-TBS)/TBS $\times$ 100%)).

obtained by aerosolizing suspensions of polystyrene latex (PSL) spheres. The PSL suspensions were prepared with ultrapure Milli-Q water and aerosolized with pressurized filtered air. The size distributions illustrate that POPS$_{TBS}$ and POPS$_{UAV}$ both correctly size the PSL particles. Particles measured in other size bins are likely due to water residuals in the PSL suspension, the tubing, or the make-up airflow, and both POPS also agree reasonably well here, across all size bins. Differences in con-

centrations measured in the 210-250 nm size bin were 3% while measuring 246 nm PSL, and differences in the 475-575 nm size bin were 8% while measuring 522 nm PSL, which again lie within the 10% uncertainty for POPS number concentrations reported by Pilz et al. (2022).

To measure supermicron particles, 3 μm polyethylene glycol (PEG) particles were generated using a Vibrating Orifice Aerosol Generator (VOAG 3450, TSI). Measurements from POPS$_{TBS}$ were compared to an Aerodynamic Particle Sizer (APS

3221, TSI), as shown in Figure B2c. The APS aerodynamic diameters were converted to volume equivalent diameters using the density of PEG of 1.125 g cm$^{-3}$ and a shape factor of 1. Furthermore, the APS data was rebinned and renormalized to match the bin widths of the POPS instrument, to make the size counts comparable. POPS$_{TBS}$ correctly sized the 3 μm PEG particles, and the concentrations in the 2585-3370 nm size bin agree with the APS concentrations within 44%, similar to the APS and POPS differences under polydisperse ambient air (see third experiment below). At this time, POPS$_{UAV}$ was not available for

experiments, but based on the previous comparisons of POPS$_{UAV}$ and POPS$_{TBS}$ in the first experiment (Fig. B1), we expect that they would perform similarly here.

Finally, in the third experiment, we compared POPS$_{TBS}$ measurements to an Aerodynamic Particle Sizer (APS 3321, TSI) and a Scanning Mobility Particle Sizer (SMPS: electrostatic classifier 3082 with CPC 3787, TSI) while measuring ambient air in a laboratory (Fig. B3). SMPS and APS sizes were converted to volume-equivalent diameters, using a shape factor of

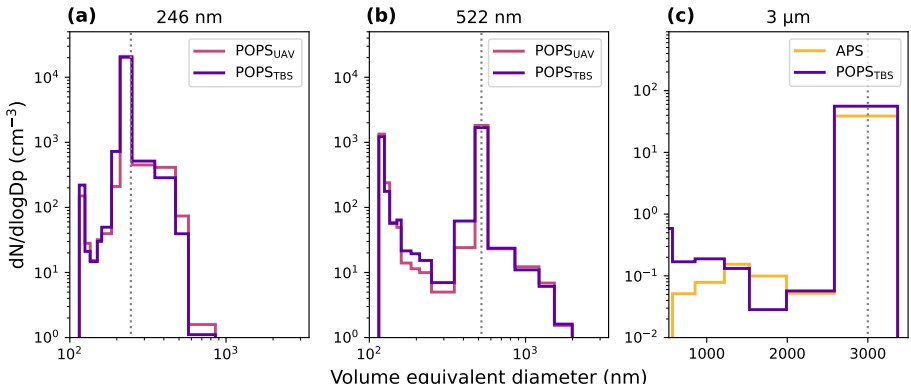

**Figure B2. (a)** and **(b)**: Size distributions from laboratory measurements of aerosolized polystyrene latex (PSL) spheres of size 246 nm (a) and 522 nm (b), measured by both $POPS_{UAV}$ (magenta) and $POPS_{TBS}$ (purple). Each size distribution represents 60 seconds of measurement. **(c)**: Size distributions from laboratory measurements of 3 μm aerosolized polyetheylene glycol, measured by $POPS_{TBS}$ (purple) and an Aerodynamic Particle Sizer (APS, yellow). Size distributions represent 90 seconds of measurement. Vertical dotted grey lines show the respective true diameters of the generated particles.

1.2 and particle density of $2\,\mathrm{g\,cm^{-3}}$, consistent with ambient air estimates (Thomas and Charvet, 2017). Similar to previous studies (Gao et al., 2016; Liu et al., 2021; Kasparoglu et al., 2022), the size distributions measured by $POPS_{TBS}$ agree well with the APS and SMPS in the overlapping size range (Fig. B3a). To allow a better comparison between the instruments, the SMPS and APS data were rebinned and renormalized to match the bin widths of the POPS instrument (Fig. B3b). Then, percent differences could be calculated for each POPS size bin, and for the total particle number concentration (sum of all bins). For particle number concentrations, $POPS_{TBS}$ measured $28 \pm 4\%$ higher concentration than the SMPS, and $44 \pm 8\%$ lower concentration than the APS. For each size bin, POPS bin concentrations were generally within 70% of the respective bins of the APS and SMPS, with the exception of two size bins with up to 120% difference.

## Appendix C: Vertical profiles of the measurement UAV in the boundary layer

Figure C1 shows vertical profiles of the particle number concentration (125 - 3370 nm size range) up to 1950 m amsl (1030 m agl) for the ascent and descent of 34 vertical profile flights of the measurement UAV (flight speed of $10\,\mathrm{m\,s^{-1}}$). The flights were conducted on 14 different days, at varying times, in February, March, and December 2022 and January and February 2023 at the main site of the CLOUDLAB project (Henneberger et al., 2023). The boundary layer height can be recognized in many of the profiles where there is a strong negative gradient in particle concentration, e.g., in subplots (h), (j), (k), (m), (ag), and (ah). In nearly all of the profiles, the ascent and descent measurements are in good agreement and closely overlap. In the two profiles on 28 January 2022 (Fig. C1a and b), the descent measurements strongly deviated from the ascent measurements, including several extreme outliers (concentration $> 5000\,\mathrm{cm^{-3}}$); we have no explanation for this, though it was likely caused

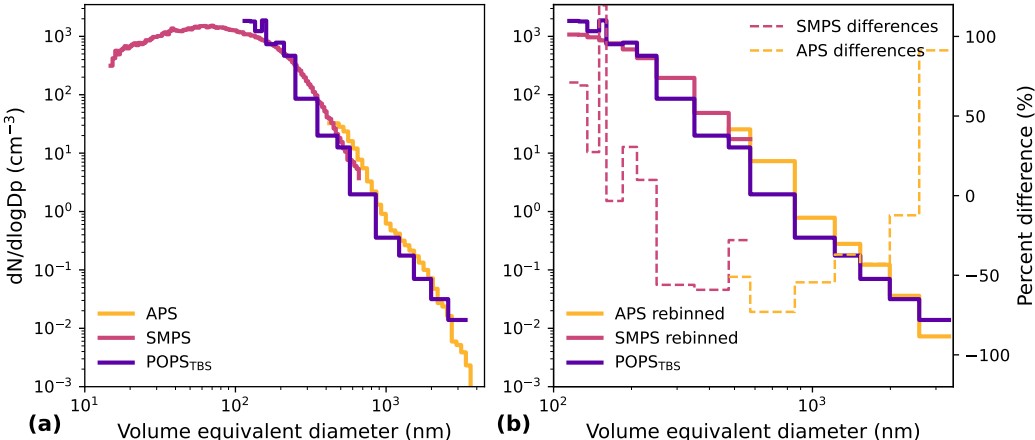

**Figure B3. (a)** Size distribution with volume equivalent diameter (nm) measured by the POPS$_{TBS}$ (purple), an Aerodynamic Particle Sizer (APS, yellow) and a Scanning Mobility Particle Sizer (SMPS, pink) in ambient laboratory air over 2.5 hours. **(b)** Similar to a), but the APS and SMPS data were rebinned and renormalized to match the bin widths of POPS. Subsequently, the percent differences between POPS and SMPS or APS were calculated for each bin (dashed pink and dashed yellow lines, respectively, with the right y-axis).

by an error in the instrument and was not reflective of the true character of the atmosphere. Similarly, there are a few other data points with unusually high concentration, e.g., in Fig. C1ag), and these data can be excluded as outliers. A quantitative comparison of the vertical ascent and descent is presented in Section 3.3.

Particle number counts considering only the supermicron particles (1220 - 3370 nm size range) are shown in Figure C2 for the same profiles as in Figure C1. For these profiles, the counts of supermicron particles are in general very low, $< 10$ particles s$^{-1}$, which means that quantitative differences are limited by counting statistics in many cases. Still, we can see that for most profiles, the supermicron counts are relatively similar for the ascent and descent. The exceptions are subplots (ad), (ae), (af), and (ag), where the ascent counts are much higher than the descent, but since this only occurs on these four profiles, we can

consider these as outliers. Overall, these profiles of total particle number (Fig. C1) and supermicron particle number (Fig. C2) indicate that both accumulation mode and coarse mode particles are sampled similarly in the ascent and descent of a flight.

*Code and data availability.*   Data and scripts available at https://doi.org/20.500.11850/640942.

*Author contributions.*   UL, ZAK, JH, FR conceived of the idea of CLOUDLAB and obtained funding. AJM, FR, ZAK, JH contributed in designing the modifications to the UAVs. AJM conducted laboratory validations of POPS. AJM and NO conducted the rotor comparison

experiments. AJM, FR, JH designed and conducted the out-of-cloud seeding experiment presented here. AJM, FR, NO, CF, RS, HZ, JH designed and conducted the in-cloud seeding experiment presented here, with conceptual input from UL and ZAK. AJM performed the data

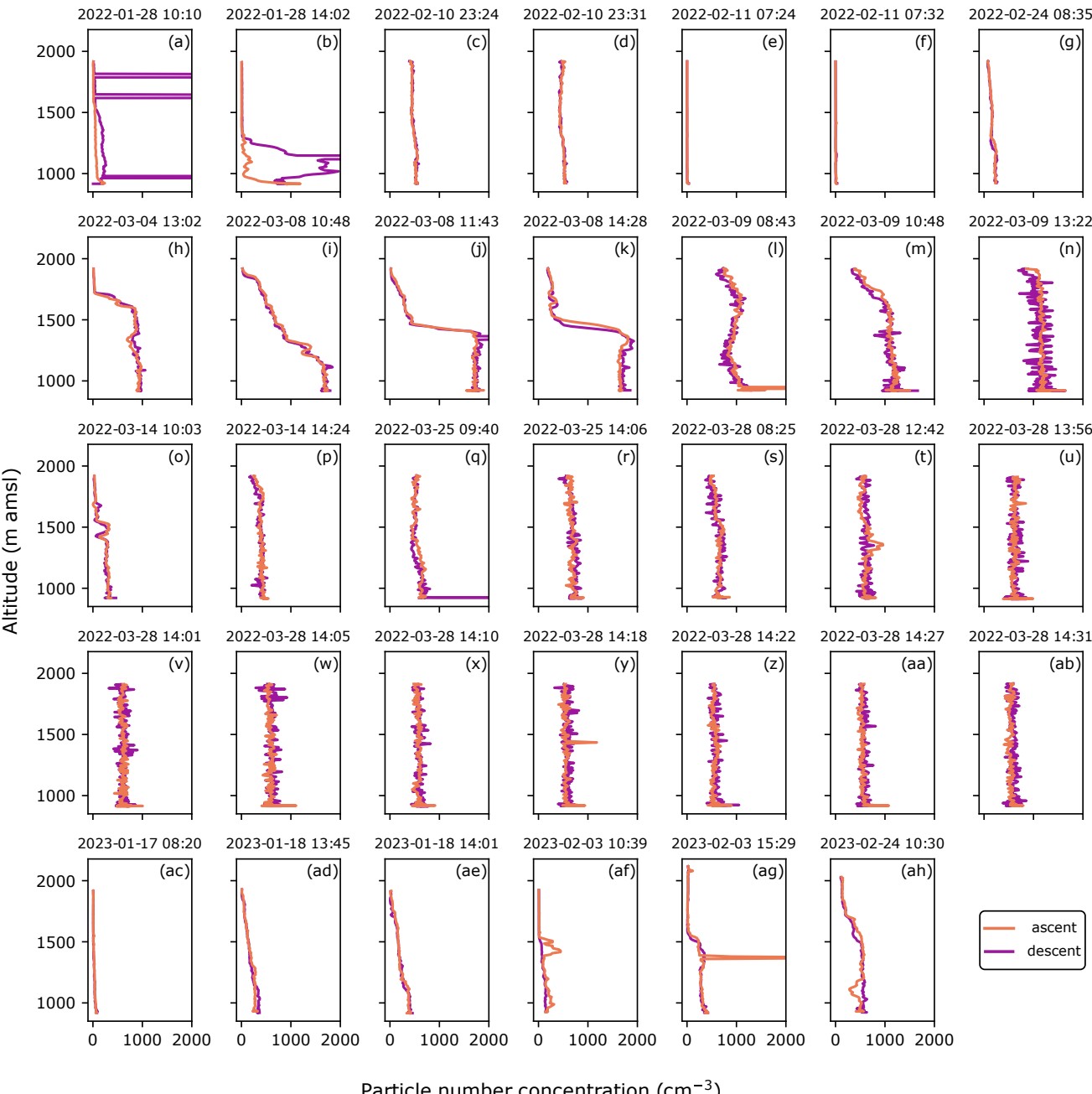

**Figure C1. (a)-(ah)** All 34 profiles to 1950 m above sea level (1030 m above ground) performed by the measurement UAV, with ascent (orange) and descent (purple) measurements of particle number concentration are shown (125 - 3370 nm size range). The start time of each profile is written above each panel.

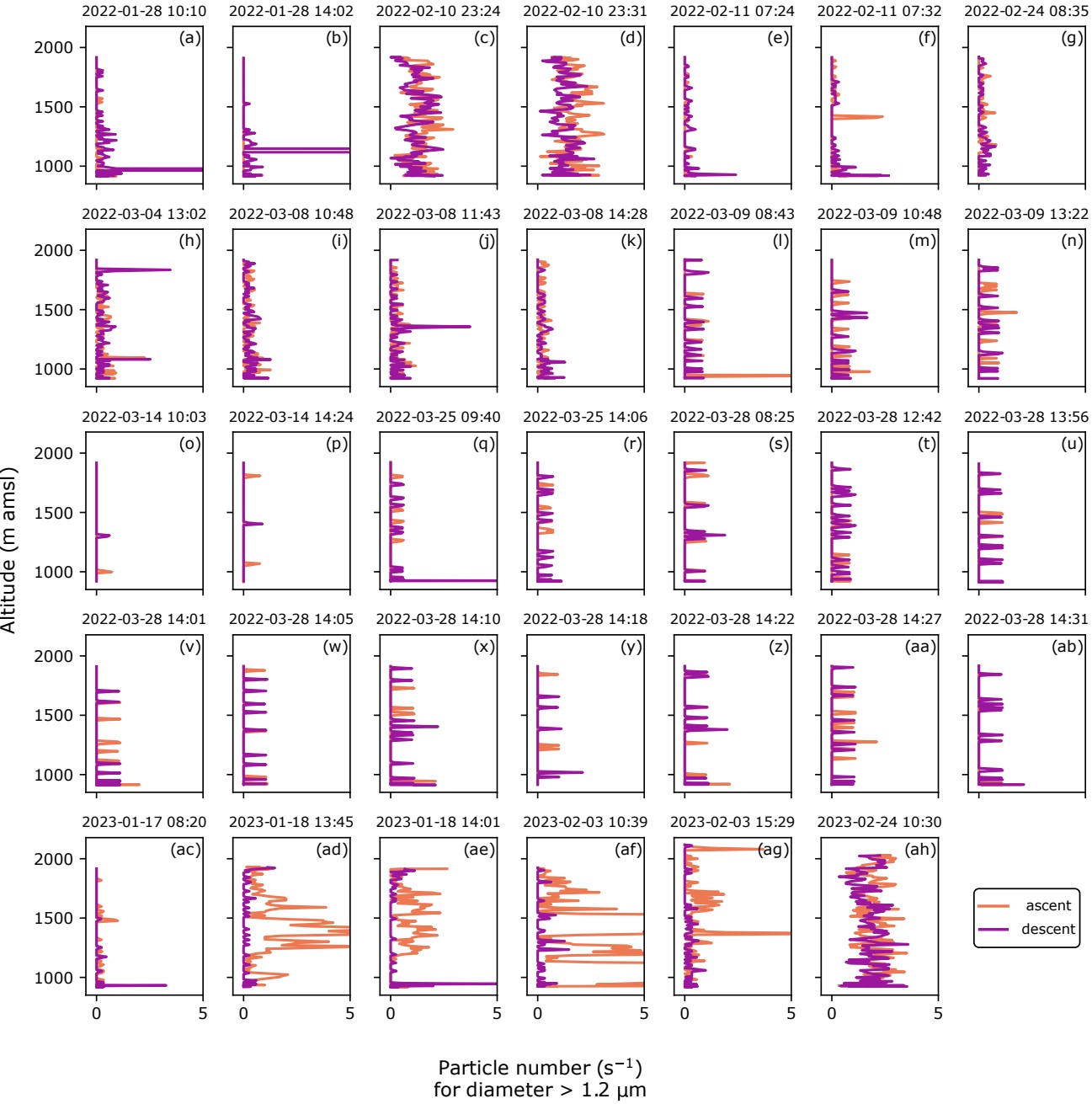

**Figure C2. (a)-(ah)** Supermicron particle number counts (1220 - 3370 nm size range) for all 34 profiles to 1950 m above sea level (1030 m above ground) performed by the measurement UAV, with ascent (orange) and descent (purple) measurements shown. The start time of each profile is written above each panel.

analysis of POPS data and created all figures presented here, except for Figure 3 created by FR. FR processed and analyzed the HOLIMO data in Figure 11. AJM wrote the manuscript. All authors contributed to the editing and review of the manuscript.

*Competing interests.* Zamin A. Kanji is a member of the editorial board of Atmospheric Measurement Techniques. The peer-review process was guided by an independent editor and the authors have no other competing interests.

*Acknowledgements.* The CLOUDLAB project has received funding from the European Research Council (ERC) 411 under the European Union's Horizon 2020 research and innovation program (grant agreement 412 No. 101021272 CLOUDLAB).

The authors would like to sincerely thank and acknowledge: Lukas Hammerschmidt, Daniel Schmitz, Philipp Kryenbühl, Remo Steiner, Dominik Brändle and the other Meteomatics employees for the interactive development of our UAVs including the flight pattern, POPS integration, and flare attachment and ignition, as well as for the valuable expertise, discussions, and prompt assistance with all things related to our Meteodrones; Patric Seifert, Johannes Bühl, Tom Gaudek, Kevin Ohneiser, and Martin Radenz from TROPOS for the remote sensing instrumentation and expertise which helped support the in-cloud seeding experiments; Maxime Hervo and Philipp Baettig from MeteoSwiss for the wind profiler integral to planning experiments; Jürg Wildi and Philip Bärtschi from v2sky for handling the applications for flight permits and Jeroen Kroese, Judith Baumann, and Santiago Llucia from the Federal Office of Civial Aviation (FOCA) for communication and support during the approval process; Michael Rösch for helping design and print the POPS inlets and for making the POPSBox for the TBS platform; Jannis Portmann for assistance in the field with drone flights; Robert David and Toni Klausen for including POPS in their VOAG experiments; Brian Rainwater from Handix Scientific for help and expertise with the POPS technical details; Frank Kasparek and Aleksei Shilin from Cloud seeding Technologies for discussions about the seeding flares; the Swiss army and the Gütergemeinde Hinterdorf Eriswil for allowing us to use their property and giving us a base; Stefan Minder for the maintenance of our base; and finally, all the farmers around Eriswil who graciously let us use their land as drone launching locations at all hours of the night and day.

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
