# Peer review of "Two new multirotor Uncrewed Aerial Vehicles (UAVs) for glaciogenic cloud seeding and aerosol measurements within the CLOUDLAB project"

_Atmospheric Measurement Techniques, 2023_

## Author Comment (AC1)

**Author's comment for the editor: Overview of changes made in the revised manuscript**

We thank the reviewers for the valuable feedback on our manuscript. We've implemented many changes, as described in detail in the responses to the reviewers, as well as a few additional changes. We present here a brief overview of the major changes to the revised manuscript:

- New section (Appendix A) presents estimates of POPS inlet sampling efficiencies, including new figure (Fig. A1)

- New section (Section 3.1) to illustrate POPS measurement uncertainties, which refers to Appendix B:

    o New figure (Fig. B1) with data from a comparison between $POPS_{TBS}$ and $POPS_{UAV}$ under ambient air

    o New subplot (Fig. B2c) showing POPS measurements of 3 μm particles

    o New subplot (Fig. B3b) showing the quantification of the differences between POPS, APS, and SMPS measurements

- Section 3.2 has been extended to include quantitative comparisons of the effect of rotors, including an updated Figure 4c with lines that show the difference between the size distributions.

- Section 3.3 has been updated to include a discussion of the coarse mode particle measurements during ascent and descent (corresponds to new Figure C2).

- Figure 5 has been updated – the particle number concentrations are now binned in altitude intervals of 20 m instead of 100 m.

- The previous appendix on the data filter for high concentration POPS data is now in the main text (Section 3.4)

- Section 4 has been restructured to present the POPS estimates of PBL height before the ceilometer PBL height

- Figure 11 has been updated to include subpanel (c), which shows timeseries of cloud droplet number concentration and ice particle number concentration from the holographic imager, taken at the same time as the POPS measurements

**Author's response to review of "Two new multirotor UAVs for glaciogenic cloud seeding and aerosol measurements within the CLOUDLAB project"**

We thank the referee for carefully reviewing our manuscript. We will address the reviewer's comments (black), present our responses (red), and highlight the changes that were made to the revised manuscript (blue). All line numbers in the author's response refer to the revised manuscript.

**Referee #1**

**General comment**

In the presented manuscript, Miller et al. showed the feasibility of initiating and observing an aerosol particle plume with uncrewed platforms that are intended for cloud seeding experiments. One of two newly developed uncrewed aerial vehicles (UAVs) handles the burning of flares used as the particle source for cloud seeds. Another UAV serves as a platform for measuring the released aerosol particles with an optical particle size spectrometer (OPSS). The major novelty of the two UAVs is their capability to be operational inside clouds and under icing conditions. An additional tethered balloon system was used as a second reference for measuring the aerosol plume created by the flares with another OPSS of the same type as the one onboard the UAV. It was successfully demonstrated that the measurement UAV and the TBS system captured the particles sourced by the seeding UAV inside and outside of clouds. The manuscript is generally well-written, has a good structure, and the figures are well-illustrated. Therefore, the work adds a valuable contribution to the atmospheric measurement community and is suitable for publication in AMT with minor revisions.

We appreciate this positive feedback and we are glad that our work was well understood.

However, more emphasis should be given to the aerosol sampling methods with the OPSS onboard the UAV and the TBS. There are generalized conclusions about the negligible impact of the UAV rotors on aerosol sampling that are not entirely demonstrated and doubtful. The two aerosol properties particle number concentration (N) and particle number size distribution (PNSD) provided by the OPSS are partly mixed, leading to conclusions that are true for one but not for the other variable. The authors need to consider that the PNSD covers several orders of magnitude in N from the smallest to the largest size bin. Losses of larger particles do not have a large effect on N but on the PNSD and its derivatives volume and mass size distribution. These derivatives are highly relevant for optical closure studies with remote sensing observations or for air quality measures. Therefore, the authors need to clearly separate and specify when speaking of measurement uncertainties. In this context, an essential characterization of inlet sampling efficiencies for varying particle diameters depending on wind speed and UAV ascent/descent rate or horizontal speeds needs to be added.

We sincerely thank the reviewer for the critical feedback and suggestions that improved our manuscript. In this revision, we have re-considered our measurement uncertainties and tried to improve our discussions by clearly referring to particle number concentrations or particle size distributions. We have also added a section to the appendix to discuss inlet sampling efficiencies for our system. We hope that our revised manuscript satisfactorily addresses the previous concerns.

**Detailed comments**

Title: the general recommendation is to avoid abbreviations in titles
We agree that the UAV should be spelled out for more clarity. However, CLOUDLAB is not an acronym and is just the name of our project, thus we have left it as is.
Adapted title: "Two new multirotor Uncrewed Aerial Vehicles (UAVs) for glaciogenic cloud seeding and aerosol measurements within the CLOUDLAB project"

Line 11 – 12: The negligible effect of rotor turbulence was not sufficiently demonstrated for particle number size distribution measurements and the statement is contradictory to Liu et al. 2021

We've adapted the sentence to be more specific to particle number concentrations rather than "aerosol measurements", and to say there is a small effect rather than a negligible effect: "In this paper, we first show validations of the POPS onboard the measurement UAV, demonstrating that the rotor turbulence has a small effect on measured particle number concentrations." (lines 11-12).

Line 17 -18: please specify here and in the following as particle number concentration and particle number size distribution

Done: "Particle number concentrations and particle number size distributions of the seeding plume…" (lines 17-18) In the remainder of the manuscript, we've adjusted our terminology to also refer specifically to particle number concentrations or particle number size distributions.

Line 24- 25: airborne observations are certainly not needed to calibrate remote sensing measurements

The text was adapted to: "In situ data can complement remote sensing measurements and are also used for initializing and validating weather prediction models…" (line 47)

Line 47: please remove the URL of Meteomatics and transfer it to the references

Done: "for the Meteodrone® (Meteomatics AG, 2023)." (line 49), with the reference "Meteomatics AG: Mobile Weather Drones, https://www.meteomatics.com/en/meteodrones-weather-drones/, 2023."

Line 52: only the year of the publications should be in brackets

The parentheses around the other words were removed:
"The first glaciogenic cloud seeding experiments were conducted in the 1940s by Schaefer (1946) using dry ice and Vonnegut (1947) using silver iodide particles,…" (lines 64-65)

Line 64: the acronym CLOUDLAB needs to be introduced

CLOUDLAB is not an acronym, it is the name of our project as it was funded by the EU (https://cordis.europa.eu/project/id/101021272). To make it clearer, we've adapted the sentence: "In our project named "CLOUDLAB", we use a multirotor UAV to seed…" (line 78)

Line 83: 90 km h$^{-1}$ is the common abbreviation used at Copernicus

We've changed it to m s$^{-1}$ according to Reviewer 2's suggestion: "They can fly for approximately 20 minutes at a maximum speed of 10 m s$^{-1}$ and can withstand wind speeds up to 25 m s$^{-1}$" (lines 99-100)

Line 113: 5 cm above rotor level seems low in comparison to other studies to avoid turbulence sufficiently. Please discuss your design in comparison with other studies and provide an analytical sampling efficiency estimation of the inlet for typical wind speed, ascent/descent rates, and horizontal velocity above ground

We agree that if the inlet were higher, as it is in Liu et al. (2021) for example, it would likely better escape rotor turbulence. However, we were still able to show that there is minimal effect on particle measurements, as discussed now more carefully in Laboratory-based POPS measurement validations. Furthermore, we've added Appendix A: Sampling efficiency of POPS inlet to discuss inlet sampling efficiencies, referenced in the text: "A detailed discussion of the inlet sampling efficiencies is given in Appendix A" (line 138). Based on our calculations, which give the conservative limit, we find that supermicron particles would have inlet sampling efficiencies approaching zero, though in reality we can measure supermicron particles (see new Figure C2, showing supermicron particle

counts in the measurement UAV profiles). Due to the complexity of our system a more accurate estimate could only be achieved with conducting computational fluid dynamics simulations.

Line 173: What means instrument variability? Uncertainty in particle number concentration measurements against reference instruments or inter-unit variability? To what time resolution does it refer, 1 Hz data? If the uncertainty refers to particle number concentration, it cannot be derived from average deviations across each size bin as presented in Appendix A1. Particle numbers at small sizes are orders of magnitude higher compared to larger sizes. Thus the uncertainty per bin has to be set in relation to the number concentration in each bin. It is better to compare integrated particle numbers across the whole size range.

We've tried to be more careful about instrument uncertainties, and we added a subsection to specifically summarize our validation tests, Section 3.1 Laboratory-based POPS measurement validations. To better test the inter-unit variability, we added an experiment comparing the two POPS under normal ambient air (Figure B1), and added to the main text: "To assess the quality of the number concentration measurements, ambient air in the laboratory was simultaneously sampled by the two POPS instruments over a 5-hour period. We found that $POPS_{TBS}$ measured a 5% lower mean particle number concentration than $POPS_{UAV}$ (Fig. B1a) and the values varied by 11% (at the 95% confidence interval) in both instruments. Thus, our results agree with those of Pilz et al. (2022), who found an uncertainty of ±10% for total number concentration. In terms of measuring particle size distributions, the two POPS are in good agreement for most size bins with counting differences below 10% (Fig. B1b). Four size bins (bins 8, 9, 11, and 12) show differences in counts up to 31%, with $POPS_{TBS}$ counting lower values than $POPS_{UAV}$." (lines 203-209). This analysis was done for the 1 Hz data. In all of the tests presented in Appendix B: Laboratory measurement validations of POPS and in Section 3.1 Laboratory-based POPS measurement validations, we've stated the differences in terms of both particle number concentration and in size bins of the particle size distributions.

Line 174-175: The method presented in Appendix A2 seems not sufficient to derive a general uncertainty of 50 % for each size bin. Please discuss the difference in your findings from other studies e.g. Liu et al. 2021 or Pilz et al. 2022

Thanks for this comment. We did not fully explain this before and we have also re-done the analysis. Figure B3 has been updated to illustrate how the calculations were performed, and the accompanying text reflects this update: "Finally, in the third experiment, we compared $POPS_{TBS}$ measurements to an Aerodynamic Particle Sizer (APS 3321, TSI) and a Scanning Mobility Particle Sizer (SMPS: electrostatic classifier 3082 with CPC 3787, TSI) while measuring ambient air in a laboratory (Fig. B3). SMPS and APS sizes were converted to volume-equivalent diameters, using a shape factor of 1.2 and particle density of 2 g cm$^{-3}$, consistent with ambient air estimates (Thomas and Charvet, 2017). Similar to previous studies (Gao et al., 2016; Liu et al., 2021; Kasparoglu et al., 2022), $POPS_{TBS}$ ambient air size distributions agree well with the APS and SMPS in the overlapping size range (Fig. B3a). To calculate percent differences between the instruments, the SMPS and APS data were first rebinned and renormalized to match the bin widths of POPS (Fig. B3b). Then, percent differences could be calculated for each POPS bin, and for the total particle number concentration (sum of all bins). For particle number concentrations, $POPS_{TBS}$ measured 28 ± 4% more than the SMPS, and 44 ± 8% less than the APS. POPS bin concentrations were generally within 70% of the respective bins of the APS and SMPS, with the exception of two bins with up to 120% difference." (lines 597-607).

The section in the main text was also modified: "In addition, the POPS measurements were compared to measurements from a Scanning Mobility Particle Sizer (SMPS) and an Aerodynamic Particle Sizer (APS). Differences in particle number concentration in the relevant sizes were 28 ± 4% compared to the SMPS and −44 ± 8% compared to the APS (Fig. B3). Differences in the size distributions were determined by rebinning the SMPS and APS data to match the respective POPS bin widths and then comparing bin concentrations: differences in bin concentrations between POPS

and SMPS and between POPS and APS were both within 70%, except for two outlier bins up to 120% (Fig. B3). However, because these three instruments have different measurement principles, comparing them unavoidably brings additional uncertainty and we cannot know the ground truth. Nevertheless, the measurements agree reasonably well, in line with similar studies by Liu et al. (2021) and Gao et al. (2016)." (lines 212-219).

Line 188-190: the turbulence created by hovering is certainly different from that created during ascent/descent or horizontal movement of the UAV, thus the conclusion seems not sufficiently supported

We've adapted this paragraph to be more specific about assessing the differences between hovering and not hovering, in addition to modifying Figure 4c to show the differences in size distributions: "When comparing the concentration differences in each size bin during the first experiment at 3 m (Fig. 4c), accumulation mode particles (120-855 nm) are on average within 10%, and coarse mode particles (>855 nm) were undercounted on average by 15% (up to 30%) when the UAV was hovering. These small differences suggest limited effects from rotors in this experiment. During the second experiment at 50 m, the hovering UAV overcounted particles in both size ranges: accumulation mode particles were on average overcounted by 22% (up to 107%) and coarse mode particles were on average overcounted by 39% (up to 44%). These differences partly arise from comparing two different POPS (whereas the previous experiment uses the same POPS in two modes), especially because the bins with the greatest discrepancies (bins 8, 9, 12, 13, and 14) are some of the bins with the largest differences in the laboratory comparison (Sect. 3.1). Nevertheless, the differences between $POPS_{TBS}$ and $POPS_{UAV}$ while hovering (up to 100%) were larger than the differences between $POPS_{TBS}$ and $POPS_{UAV}$ measured during the laboratory experiments (up to 30 %) (Sect. 3.1). This is most likely due to effects from the UAV rotors. Therefore, we add additional uncertainties of ±22% for accumulation mode particles and ±40% for coarse mode particles for $POPS_{UAV}$ while flying or hovering. However, the differences in mean total particle number concentration were still below 5% for both experiments, indicating that the rotor-induced turbulence has little effect on the total particle number concentration." (lines 234-246)

Line 194-195: the difference are hard to derive on a logarithmic scale, but it seems that deviations between sampling with and without rotors near the inlet are up to a factor of two for particle diameters above 800 nm. This could become relevant when deriving particle properties from the number size distribution, e.g. volume size distribution for optical closure studies e.g. Düsing et al. 2021 (https://doi.org/10.5194/acp-21-16745-2021)

Thanks for this comment. We agree that it is important to be more transparent about the effects of the rotors. We've adapted Figure 4c: we show the same size distributions with added lines to highlight the differences between the distributions. We've also adapted the text to be more specific: see response to the above comment.

Line 201: ascending/descending rates of 10 ms$^{-1}$ can significantly reduce sampling efficiency of supermicron particles depending on the inlet geometry. Please provide an analytical estimate

Yes, it is true that the ascending/descending flight speeds significantly reduce sampling efficiency, and we added the new section Appendix A: Sampling efficiency of POPS inlet to discuss this. We've also added a new figure (Fig C2) and a paragraph to discuss specifically the differences in ascent and descent for supermicron particles: "Because the total particle number concentration is dominated by the high number of accumulation mode particles in comparison to coarse mode particles, we also compared the difference in coarse mode particles in the ascent and descent of all the profiles (Fig. C2). We would expect measurements of coarse mode particles to be more affected by the UAV flight rather than accumulation mode particles, because small particles generally follow the streamlines of the air flow, whereas large particles have more inertia and can deviate from the streamlines. Therefore, we might expect an enhancement of coarse mode particles in the ascent and

a depletion in the descent because the inlet is pointed upwards. However, the coarse mode particle number concentrations in the ascent and descents are very similar, with the exception of four profiles where the ascent does have higher particle counts (Fig. C2). A quantitative assessment is limited by the fact that there are so few coarse mode particles measured: in nearly all profiles, coarse particle counts are less than 10 particles s$^{-1}$. The low number results from the low presence of coarse particles in the atmosphere and may be additionally reduced due to our general limited sampling efficiency of supermicron particles during flight in either direction (discussed in Appendix A)." (lines 273-284).

Line 218: rotor downwash does not appear neglectable in general, it rather depends on (1) averaging altitude range and (2) ascending/descending rates. Also, (3) the variations in particle number concentration in Fig. B1 appear somewhat higher during descent than during ascent, thus indicating effects from downwash

(1) Using an averaging altitude range of 20 m instead of 100 m, the trend still holds that the particle number concentrations of the ascent and descent agree very well (y=0.97+29, r$^2$=0.90). Since we have a sampling rate of 1 Hz and a flight speed of 10 m s$^{-1}$, the sampling resolution is approximately 10 meters, therefore an averaging interval of 20 meters is appropriately small. Thus, we would argue that any effects from the rotor downwash are negligible when looking at particle number concentrations. We have changed Fig. 5 in the manuscript to show the altitude averaging interval of 20 m, and changed the text appropriately: "…the particle concentration measurements were first binned into altitude intervals of 20 m and then averaged over each interval on the ascent and the descent of each flight." (lines 262-263)

(2) We acknowledge that the downwash and turbulence depend on the ascending/descending flight speed, but we have a fixed ascending/descending flight speed of 10 m s$^{-1}$ and thus cannot test those effects. We have discussed these effects briefly in the new section Appendix A: Sampling efficiency of POPS inlet.

(3) It is apparent that the variations in particle number concentration in Fig C1 (previously Fig B1) are slightly larger in the descent than in the ascent, which can indicate some effects of rotor turbulence. However, since the mean particle number concentrations are statistically similar even over small altitude averaging ranges, we do not consider this to be a significant problem. We do already acknowledge this in the text, but have modified the text to make it more clear:

"Often the descent flight measurements have more variability than the ascent flight measurements (e.g., in Fig. B1n), likely due to influences of rotor turbulence or flight instabilities during descent. However, as can be seen in the quantitative assessment described below, this does not significantly affect the mean concentrations, even over small averaging intervals." (line 257-260)

Line 219-225: that appears as a valuable extension of the POPS usage and should therefore be part of the main manuscript
Thanks for this suggestion. Accordingly, we have now moved it to the main part – see new Section 3.4 Data quality filter for POPS measurements at high concentrations (lines 285-349)

Fig. 6 (b): please specify particle number concentration on x-axis
The x-axis in Fig 7b (prev. 6b) has been changed to "Particle number concentration (cm$^{-3}$)"

Line 242: please briefly discuss the (1) uncertainty of the POPS, e.g. relation of sampling frequency of 1 Hz and ascent/descent rate of 10 ms$^{-1}$ including measurement uncertainty for 1 Hz data. (2) Is the ascent/descent rate appropriate for profiling? (3) Would it be useful to average ascent and descent

profiles of particle number concentration to one profile like for temperature and humidity (which are sampled at 10 Hz)

(1) The sampling resolution of POPS in a vertical profile is 10 m, considering the 1 Hz sampling frequency and the 10 m s$^{-1}$ flight speed. There is further uncertainty from the GPS-based altitude measurement of 20 m. We've changed the sentence in the text to be more clear with what we mean: "The PBL heights derived from the POPS measurements are associated with an uncertainty of ±20 m because 1) the sampling frequency of POPS (1 s$^{-1}$) multiplied by the flight speed (10 m s$^{-1}$) gives a sampling resolution of 10 m, and 2) the GPS altitude measurements have an estimated uncertainty of 10 m. Therefore, the POPS-derived PBL heights from the ascent and descent are in good agreement with each other and with the RH-derived PBL height." (lines 361-366)

(2) A slower ascent/descent flight speed would improve the sampling resolution and thus could be more appropriate for profiling if the goal is to be more precise than 10 m. However, because the temperature, RH, and wind measurements are optimized and validated only for a flight speed of 10 m s$^{-1}$ as this is the operational vertical flight speed for Meteodrones to maximize the reachable altitude. Therefore, we did not test other flight speeds here. We have added this information to the text: "All meteorological measurements are validated and calibrated by the manufacturer for the operational profiling flight speed of 10 m s$^{-1}$." (lines 107-108).

(3) Yes, it could be helpful to average together the ascent and descent. For Figure 7, we want to show both the ascent and descent to illustrate how they compare to each other.

Line 241: can the measurement UAV really characterize a ceilometer? It is rather a complementary method
We change the sentence to: "…the measurement UAV can characterize the lower atmosphere similarly to a ceilometer, with the advantage that.." (lines 374 - 375)

Fig. 9: please specify particle number concentration on y-axis
The y-axis in Fig 10 (prev. 9) has been changed to "Particle number concentration (cm$^{-3}$)"
In addition, the "concentration" axis labels in Fig 6 (prev. C1), Fig 11a (prev. 10a), and Fig B1 have been changed to "particle number concentration".

Fig 10 (a): the plot would benefit from an additional line showing the time-height series of the balloon with the height scale on the right y-axis
Thanks for the suggestion, and we have subsequently added a grey line for the altitude of the TBS on the right y-axis of Figure 11c (prev. 10a). The altitude line is referenced in the text: "After this seeding mission, the TBS was brought back to ground very soon after the experiment (see altitude of TBS, Fig. 11b), thus allowing measurements of three different conditions in a short period of time:…" (lines 436-439). Note that we have also added timeseries of cloud droplet and ice crystal number concentrations from the HOLIMO measurements, also shown in Figure 11b.

Line 319: The cloud droplets appear quite small if they are measured by the POPS. Are there measures that support this hypothesis, e.g. cloud droplet diameter derived from the cloud radar, or relative humidity measurements of the POPS sample stream
We are certain that we were measuring within the cloud based on measurements from cloud radar and HOLIMO , i.e. the presence of cloud droplets and ice crystals (now shown in Figure 11b). The lower size limit of HOLIMO is 6 μm, meaning, we do not have an additional measurements to validate the small droplet sizes. However, previous studies have shown that continental stratus clouds often have cloud droplets as small as a few micrometers (Miles et al., 2000). Furthermore, we hope it is clear in the text that we are hypothesizing that they are cloud droplets based on the differences between the size distributions. This of course is  based on the size measurement as we

are not trying to be quantitative about the number of cloud droplets. We also acknowledge that if they are cloud droplets, they would certainly be at the smaller end of the droplet size spectrum.

Line 326: please provide additional information on the inlet of the POPS $_{TBS}$ . Airborne particle measurements should assure sample air relative humidity below 40 % to provide the dry particle diameter by heating or using a dryer
Neither of our POPS has a drying mechanism, thus our particle diameters are humidity-dependent. We've added to Section 2.2 to clarify this: "The sampled particles are not dried prior to measurement, thus POPS$_{UAV}$ reports particle diameters that are humidity-dependent and have to be interpreted along with the relative humidity measured by the Meteodrone sensor." (lines 136-137), and reference that in Section 2.5:  "POPS$_{TBS}$ has an inlet design identical to that of POPS$_{UAV}$ (see Section 2.2)." (line 194).

Line 334-335: this conclusion is not sufficiently supported
We have adapted the sentence to be more appropriate: "We then showed that the POPS data are comparable to other aerosol instrument measurements (particle number concentrations within 50%; Sect. 3.1) and that there is a minimal effect of rotor-induced turbulence from the UAV on particle number concentration (Sect. 3.2 and 3.3)." (lines 475-477)

Line 339: particle number size distributions during profiling were not shown
That's true. We have changed the phrase: "First, the measurement UAV can be used for profiling the atmosphere, i.e., measuring temperature, humidity, wind, and particle number concentrations." (lines 481-482)

Line 344: without proper estimations of uncertainties for particle number size distribution measurements and information about the sample air relative humidity, the system can certainly not serve as a "benchmark" to validate remote sensing retrievals
We have replaced this part of the sentence: "Profiling the atmosphere with in situ measurements is important for understanding and predicting local air quality and health effects, atmospheric transport, and boundary layer meteorology, for which our measurement UAV is a useful tool." (lines 486-487).

Line 361: it was not shown that microphysical changes within the plume can be assessed
It is true we have not shown that in this work. We intended it to mean that it would be possible in the future. We have clarified the sentence as follows: "We have shown that not only can we produce a cloud seeding plume from a multirotor UAV, but we can also detect seeding particles and ice crystals up to 3000 m downstream (Sect. 5), and in future work we can therefore assess the microphysical changes within the plume." (lines 502-504)

Line 371: PSL suspensions (not solutions) should only be produced with distilled water
The suspensions were prepared with ultrapure Milli-Q laboratory water and aerosolized with filtered air, but still, there were some other sized particles measured. These could be from the water, airflow, agglomerated particles, or from the tubing or material used in the experiment. We did not use a DMA to filter out these particles. Nonetheless, the particle counts of the 246 and 522 nm particles are orders of magnitude higher than the other particle counts, and both POPS agree well across most size bins. We have adapted the text as follows: "The submicron particles of 246 nm (Fig. B2a) and 522 nm (Fig. B2b) were obtained by aerosolizing suspensions of polystyrene latex (PSL) spheres. The PSL suspensions were prepared with ultrapure Milli-Q water and aerosolized with pressurized filtered air. The size distributions illustrate that POPSTBS and POPSUAV both correctly size the PSL particles. Particles measured in other size bins are likely due to water residuals in the

PSL suspension, the tubing, or the make-up airflow, and both POPS also agree reasonably well here, across all size bins." (lines 581-584)

Line 377-378: The comparison to the findings by Pilz et al. 2022 is methodologically wrong. They reported comparisons of total particle number concentration measured by a POPS against a reference CPC for specific PSL sizes filtered by a DMA and not the uncertainties in single size bins measured at undefined aerosols. Pilz et al. concluded average uncertainties for particle number concentration in the range of Gao and Mynard and not up to 110%.
We apologize for misinterpreting Pilz et al. (2022). We have removed that sentence. In our reanalysis of the uncertainties of our POPS, we have referred to Pilz et al. (2022) differently: "We found that POPS$_{TBS}$ measured a 5% lower mean particle number concentration than POPS$_{UAV}$ (Fig. B1a) and the values varied by 11% (at the 95% confidence interval) in both instruments. Thus, our results agree with those of Pilz et al. (2022), who found an uncertainty of ±10% for total number concentration." (lines 205-207)

Line 385: how are the corresponding particle diameters are selected from the SMPS and APS? The geometric centers of each bin are certainly different across the instruments
We extended our description of how we made the comparison. Additionally, we have since redone the analysis and explain it accordingly: "To allow a better comparison between the instruments, the SMPS and APS data were rebinned and renormalized to match the bin widths of the POPS instrument (Fig. B3b). Then, percent differences could be calculated for each POPS size bin, and for the total particle number concentration (sum of all bins)" (lines 602-605).

Line 407: was the flow rate of 0.9 cm³ s$^{-1}$ used for all measurements? This very low flow has certainly an effect on the sampling efficiency of supermicron particles
No, often 3 cm³ s$^{-1}$ was used. The sampling flow rate does make a difference on sampling efficiency and we discuss this in the new section Appendix A: Sampling efficiency of POPS inlet, specifically: "If we consider a lower inlet flow rate of 0.9 cm$^3$ s$^{-1}$, which we also sometimes used for POPS$_{UAV}$, then the aspiration efficiency would again approach 0% in this last considered case." (lines 554-555).
We also added a sentence in Section 2.2 to explicitly state what flow rates we use: "Flow rates of 3 cm$^3$ s$^{-1}$ or 0.9 cm$^3$ s$^{-1}$ were used for POPS$_{UAV}$. [...] A detailed discussion of the inlet sampling efficiencies is given in Appendix A." (lines 135, 138)

References:

Miles, N. L., Verlinde, J., and Clothiaux, E. E.: Cloud Droplet Size Distributions in Low-Level Stratiform Clouds, Journal of the Atmospheric Sciences, 57, 295–311, https://doi.org/10.1175/1520-0469(2000)057<0295:CDSDIL>2.0.CO;2, 2000

---

## Author Comment (AC2)

**Author's comment for the editor: Overview of changes made in the revised manuscript**

We thank the reviewers for the valuable feedback on our manuscript. We've implemented many changes, as described in detail in the responses to the reviewers, as well as a few additional changes. We present here a brief overview of the major changes to the revised manuscript:

- New section (Appendix A) presents estimates of POPS inlet sampling efficiencies, including new figure (Fig. A1)

- New section (Section 3.1) to illustrate POPS measurement uncertainties, which refers to Appendix B:

   o New figure (Fig. B1) with data from a comparison between $POPS_{TBS}$ and $POPS_{UAV}$ under ambient air

   o New subplot (Fig. B2c) showing POPS measurements of 3 µm particles

   o New subplot (Fig. B3b) showing the quantification of the differences between POPS, APS, and SMPS measurements

- Section 3.2 has been extended to include quantitative comparisons of the effect of rotors, including an updated Figure 4c with lines that show the difference between the size distributions.

- Section 3.3 has been updated to include a discussion of the coarse mode particle measurements during ascent and descent (corresponds to new Figure C2).

- Figure 5 has been updated – the particle number concentrations are now binned in altitude intervals of 20 m instead of 100 m.

- The previous appendix on the data filter for high concentration POPS data is now in the main text (Section 3.4)

- Section 4 has been restructured to present the POPS estimates of PBL height before the ceilometer PBL height

- Figure 11 has been updated to include subpanel (c), which shows timeseries of cloud droplet number concentration and ice particle number concentration from the holographic imager, taken at the same time as the POPS measurements

**Author's response to review of "Two new multirotor UAVs for glaciogenic cloud seeding and aerosol measurements within the CLOUDLAB project"**

We thank the referee for carefully reviewing our manuscript. We will address the reviewer's comments (black), present our responses (red), and highlight the changes that were made to the revised manuscript (blue). All line numbers in the author's response refer to the revised manuscript.

Referee #2

**General Comments:**

In the presented manuscript, Miller et al. describe two Uncrewed Aerial Vehicles (UAVs) for cloud research: (i) the "measurement" UAV equipped with an optical particle counter for measuring particle size between 0.1 and 3.4 μm (in diameter) and met sensors, and (ii) the "seeding" UAV equipped with seeding flares, initiating ice particles in supercooled clouds. There is a comprehensive description of the cloud seeding procedure. Tests were made to validate the POPS measurements on-board the multi-rotor, by (i) comparing the ascent and descent profiles, and (ii) assessing the observations with and without the rotors. Characterization of a dispersion of an out-of-cloud seeding plume produced by the flares on-board the UAV was made successfully through the experiment. Results are also shown from an in-cloud seeding experiment, where the seeding UAV was inside a supercooled cloud and the TBS system was downstream. This study successfully demonstrates the capabilities of the measurement UAV and the TBS system to capture the plume produced by the seeding UAV inside and outside the clouds. This work adds a valuable contribution to the atmospheric measurement community and is suitable for publication in AMT with minor revisions. We sincerely thank the reviewer for the time and care taken for reviewing this manuscript, and for the positive and constructive feedback provided. We are glad that you see the value in our work.

Although, there are some specific areas in the manuscript that need further elaboration/clarification. Here are the main parts that should be revised.

- First of all, the scope of each experiment should be clearly described in the beginning of each related section. There is no clear distinction between the technical purpose and the scientific aim of each experiment.

Thanks for this feedback. We have added three sentences to the beginning of Section 5 to explain better the purpose of the seeding experiments: "Next, we demonstrate how the seeding and measurement UAV are deployed within the CLOUDLAB project (Henneberger et al, 2023). First, we show how the measurement UAV with POPSUAV can be used to characterize the dispersion of an out-of-cloud seeding plume (Section 5.1 and Fig. 8a). The purpose of the out-of-cloud seeding experiment was to estimate the concentration and dispersion of the particles produced from the flares onboard the seeding UAV. Second, we present an in-cloud seeding experiment in a supercooled stratus cloud where the changes in the aerosol and microphysical properties induced by the seeding UAV were measured downstream by the TBS (Section 5.2 and Fig. 8b). The in-cloud seeding experiment was designed to induce ice nucleation and observe ice crystal growth in supercooled clouds. The examples presented here demonstrate the capabilities of the UAVs and other instrumentation, and further results will come in future publications." (lines 378-386)
We also added a sentence to Section 3.3 to give more context to our vertical profile flights with the measurement UAV: "The flights were performed to measure temperature, humidity, wind, and aerosol to plan our seeding experiments (see Section 5), but the flight data can also be used to assess the effect of flight on particle sampling." (line 251-253).

- The authors did not use the UAV observations to estimate the PBL height, but the UAV data were only used for validating the PBL height retrieved by the ceilometer measurements. I would

suggest that the authors describe the method of how the PBL height can retrieve only using the met profiles from the UAV, and then validating it with the results from the ceilometer.

Thanks for this suggestion – we agree that reordering the section is sensible. We have adapted the section so that it starts with deriving the PBL from the RH and particle measurements from the UAV, and then comparing those PBL estimates to those from the ceilometer. Please see Section 4 for the updated text.

- In addition, as there was a holographic imager on-board the UAV, it would be good to show some results from it.

The holographic imager was onboard the TBS. We agree that showing these data strengthens the paper and added the cloud droplet and ice crystal number concentrations from the holographic imager to Figure 11c.

- It would be good to show an in-depth characterization of inlet sampling efficiencies for a range of particle sizes and how the 3-D wind affects its efficiency, as well as whether this efficiency differs between ascent and descent profiles.

We have added a new section to discuss the inlet sampling efficiencies, Appendix A: Sampling efficiency of POPS inlet, as also suggested by Reviewer 1.

**Specific Comments:**

Lines 98, 99: It would be good to stick on the same metric system. Ms$^{-1}$, and kmh$^{-1}$ are both used.

90 km h$^{-1}$ was replaced with 25 m s$^{-1}$: "They can fly for approximately 20 minutes at a maximum speed of 10 m s$^{-1}$ and can withstand wind speeds up to 25 m s$^{-1}$" (lines 99-100)

Lines 100-105: Which met sensors are used on-board? Are they custom-made or commercial? Provide details.

The company manufacturer of the met sensors is considered intellectual property of Meteomatics and thus cannot be disclosed. However, we have added details about the types of sensors: "The standard version of the Meteodrone is equipped with sensors to measure temperature (±0.1 K; Integrated Circuit temperature sensor), relative humidity (±1.8% at 23 °C between 0-90% RH; capacitive sensor with humidity-permeable cover layer), and pressure (±1.5 hPa; Piezo-resistive sensor), as well as a calibrated system for measuring wind speed (± 1 m s$^{-1}$) and wind direction (± 10°), each at 10 Hz sampling frequency (Meteomatics, personal communication; Hervo et al., 2023)." (lines 102-106)

Line 128: Is it isokinetic inlet? Elaborate if yes, or not.

No, our inlet is not isokinetic. We have added that to the text here: "An inlet extension was designed so that the inlet (2 mm inner diameter, not isokinetic) extends out of the housing,…" (line 133). Furthermore, we have added a new section Appendix A: Sampling efficiency of POPS inlet to discuss inlet sampling efficiencies.

Lines 151-155: Needs elaboration here – provide details on how you decide the ideal seeding altitude, which parameters/conditions are important for this?

We do already list the variables which are important for us: wind, temperature, cloud altitude, temperature, and cloud structure. We have added an additional sentence to provide an example of what we look for specifically when we are going for an in-cloud seeding experiment to produce ice: "For example, when we plan an in-cloud seeding mission, during which we expect to nucleate and measure ice crystals, we target stable low stratus clouds with cloud temperatures below -5 °C (cold enough for ice nucleation to occur with silver iodide particles), low radar reflectivity (i.e., low background ice content), cloud base between 1100 and 1600 m amsl (low enough to be reached

with our UAVs and tethered balloon), and wind speeds of 3-15 m s$^{-1}$ (high enough to get advection of the seeding plume, and low enough to have safe conditions for flight of UAV and balloon)." (lines 159-164)

Line 187: Why only those 2 "small" sizes were chosen for the calibration? How about the accuracy for larger sizes than these (i.e. for 2 μm, 3 μm)?
We only showed these two measurements because they were done with both POPS, allowing for direct comparison. We also had an experiment of POPS$_{TBS}$ with the APS measuring 3 μm polyethylene glycol aerosol particles, produced using a Vibrating Orifice Aerosol Generator (VOAG, TSI), now added as a subplot Figure A1. Unfortunately, the POPS$_{UAV}$ was not available at that time to be included in the comparison. However, we do expect that the two POPS would measure similarly to each other, as they do for ambient air (also added, see Figure A1) and for the 246 and 522 nm particles. The following text was added, in addition to the new Figure A2: "To measure supermicron particles, 3 μm polyethylene glycol (PEG) particles were generated using a Vibrating Orifice Aerosol Generator (VOAG 3450, TSI). Measurements from POPS$_{TBS}$ were compared to an Aerodynamic Particle Sizer (APS540 3221, TSI), as shown in Figure B2c. The APS aerodynamic diameters were converted to volume equivalent diameters using the density of PEG of 1.125 g cm$^{-3}$ and a shape factor of 1. Furthermore, the APS data was rebinned and renormalized to match the bin widths of the POPS instrument, to make the size counts more comparable. POPS$_{TBS}$ correctly sized the 3 μm PEG particles, and the concentrations in the 2585-3370 nm size bin agree with the APS concentrations within 44%, similar to the APS and POPS differences under polydisperse ambient air. At this time, POPS$_{UAV}$ was not available for experiments, but based on the previous comparisons of POPS$_{UAV}$ and POPS$_{TBS}$, we expect that they would perform similarly here." (lines 588-596).

Line 217: Have you checked this in lower ascending/descending speeds than 10ms$^{-1}$? It would be interesting to see whether similar results are derived.
Early on in the development of the measurement UAV, we tested profiling speeds of 1, 3, and 10 m s$^{-1}$, though we were only allowed to fly to 100 m at that time. In those very preliminary tests, no differences were found between flight speeds. To optimize time and the limited use of battery, we decided to follow the Meteomatics operational flight speed of 10 m s$^1$ for our vertical flying speed, especially because the meteorological measurements are validated and calibrated for this flight speed (corresponding information was added: "… and all meteorological measurements are validated and calibrated by the manufacturer for the operational profiling flight speed of 10 m s$^{-1}$" lines 107-108). Therefore, we did not perform further profiling with other flight speeds. However, we do agree that it would be interesting to see whether similar results to our ascent/descent analysis would be found with other flight speeds.

Line 234: The ascent/descent speed for the experiment were 10ms$^{-1}$?
Yes, exactly.

Line 246-248: There are many more ways to calculate the vertical profiles of aerosol concentrations. The sentence needs rephrasing, to be completely valid.
That is true. However, in restructuring the section, as suggested in your general comment above, this sentence is now omitted.

Line 251: Text needs to be ahead the figure (i.e. Fig. 6).
We have moved the figure to be after the text. The final formatting will be done by the journal.

Line 253: How the blue line was derived by the RH minimum gradient? Provide justification.
The gradient of the relative humidity curve was calculated, and the point at which this gradient was smallest, i.e. the most negative, was taken as the height of the PBL. This is a common method for

determining PBL height, described for example in Seidel et al (2010) or Collaud Coen et al (2014). We have adapted the text to read: "The height of the PBL using this RH profile can be estimated by finding the minimum (i.e., most negative) in the gradient of RH with respect to altitude (Seidel, et al.., 2010; Collaud Coen, et al., 2014), which results in a PBL height of 1421 m amsl." (lines 357-359)

Line 302-304: Not clear aim of this experiment (Sec. 5.2). Which was the main purpose of this specific experiment? Why not using a sensor for measuring the large particles in the plume too, so as to measure the supercooled cloud droplets and ice crystals there? Was the sole scope to measure only the remaining inactivated seeding particles?

We can see how this may be confusing. The overall goal of the in-cloud seeding experiment was to produce ice crystals. We have done many of these experiments in the course of our CLOUDLAB project, and the results in terms of ice crystal production are discussed in the CLOUDLAB introduction paper by Henneberger et al. (2023). We did decide to additionally add data from the holographic imager to show cloud droplets and ice crystals in Figure 11c. We've added two sentences at the beginning of Section 5 to explain better our goals for these experiments: "Second, we present an in-cloud seeding experiment in a supercooled stratus cloud where the changes in the aerosol and microphysical properties induced by the seeding UAV were measured downstream by the TBS (Section 5.2 and Fig. 8b). The in-cloud seeding experiment was designed to induce ice nucleation and observe ice crystal growth in supercooled clouds. The examples presented here demonstrate the capabilities of the UAVs and other instrumentation, and further results will come in future publications." (lines 382-386)

Line 305: It would be good to show some results from the holographic imager.
We agree! We have added to Figure 11c timeseries of cloud droplet number concentration and ice crystal number concentration and included some mentions of it in the text:
"The seeding signal is also visible in the ice crystal number concentrations, which increase from 0 up to 500 L$^{-1}$ (Fig. 11c) at the same time as the particle number concentration increases." (lines 440-442)
"Therefore, we believe the elevated concentrations that POPS$_{TBS}$ and HOLIMO measured are the seeding plume passing by, and not natural variation in the cloud" (lines 446-447).
"We have shown that not only can we produce a cloud seeding plume from a multirotor UAV, but we can also detect seeding particles and ice crystals up to 3000 m downstream (Sect 5.2),.." (lines 502-503)

Line 311: Are there specific criteria to choose the seeding altitude?
We choose the seeding altitude based on the criteria mentioned in Section 2.4. If we have access to the whole cloud, as was the case in this experiment, then we tended to choose based on temperature. It is also a matter of logistics such as how high the TBS can fly, how bad the icing conditions are for the UAV and the TBS, etc.

Lines 337-339: Good thinking and reasoning, are there any relevant references from literature?
Thanks, and yes we have now added two references, Flossmann and Wobrock (2010) and Ohata et al. (2016) to the sentence: "..showing the effects of particle activation into cloud droplets as well as scavenging of aerosol particles by cloud droplets, as previously documented by others (e.g., Flossmann and Wobrock, 2010; Ohata et al., 2016)" (lines 561-462). The first is a review of theoretical and modeling concepts regarding wet scavenging of aerosol, and the second is recent experimental evidence for wet scavenging.

Line 341-344: Additional information is needed for the inlet of POPS. Also, could you also use any "drying and/or heating mechanism" to tackle the tackle the built-up moisture in the inlet?

Additional information on the POPS inlet was added to Section 2.2: "The sampled particles are not dried prior to measurement, thus POPS$_{UAV}$ reports particle diameters that are humidity-dependent and have to be interpreted along with the relative humidity measured by the Meteodrone sensor." (lines 136-137), and reference that in Section 2.5: "POPS$_{TBS}$ has an inlet design identical to that of POPS$_{UAV}$ (see Section 2.2)." (line 192). Further information on the inlet sampling efficiencies is added in the new section Appendix A: Sampling efficiency of POPS inlet.

Regarding a drying or heating mechanism, that would be something that could be added in the future which would help to overcome this problem. A sentence has been added for that: "For future projects, it could be worthwhile to build an inline drying or heating mechanism in the inlet, with the consequent exclusion of cloud droplets measurements due to their evaporation." (lines 469-470).

Line 350: "comparable to other aerosol instrument measurements": add some quantitative results here, e.g. XX% deviation, etc.

We added a quantification to the text: "We then showed that the POPS data are comparable to other aerosol instrument measurements (particle number concentrations within 50%; Sect. 3.1) and that there is a minimal effect of rotor-induced turbulence from the UAV on particle number concentration (Sect. 3.2 and 3.3)." (lines 474-476)

Line 355: Not correct wording. The concentration of the UAV profile was not compared to the backscatter signal from the ceilometer; as the parameter shown was not the same to be able to compare them. The sentence needs rephrasing.

Since we have restructured the section about the use case for determining the PBL, we feel that this sentence in the conclusions is now appropriate as written.

Line 376: There was not robust assessment of the microphysical properties/changes.

It is true we have not shown that in this work. We intended it to mean that it would be possible in the future. We've clarified the sentence as follows:

"We have shown that not only can we produce a cloud seeding plume from a multirotor UAV, but we can also detect it up to 3000 m downstream (Sect. 5), and in future work we can therefore assess the microphysical changes within the plume." (lines 502-503)

Line 473: XX needs to be replaced with the corresponding text.

Yes, thanks for the reminder. The data and scripts are now available at https://doi.org/20.500.11850/640942

**Figures:**

General comment: Text introducing figures should always be ahead the figures

We did this where it was possible, but we also wanted to optimize the space in the document. We trust that the final formatting will be done by the journal accordingly.

Figure 2: Not very clear image

We would like to have a higher-resolution image of the seeding UAV in-action, but this was limited given that the UAV must be at least 100 meters above us for seeding.

Figure A1: Not very clear. It would be better to visualise the two sizes in 2 different figures, one next to the other.

Thanks for this comment. We have split up the two sizes into two subplots, and added a third subplot to include measurements of 3 μm particles, as mentioned above.

**References**

Collaud Coen, M., Praz, C., Haefele, A., Ruffieux, D., Kaufmann, P., and Calpini, B.: Determination and Climatology of the Planetary Boundary Layer Height above the Swiss Plateau by in Situ and Remote Sensing Measurements as Well as by the COSMO-2 Model, Atmospheric Chemistry and Physics, 14, 13 205–13 221, https://doi.org/10.5194/acp-14-13205-2014 , 2014.

Flossmann, A. I. and Wobrock, W.: A Review of Our Understanding of the Aerosol–Cloud Interaction from the Perspective of a Bin Resolved Cloud Scale Modelling, Atmospheric Research, 97, 478–497, https://doi.org/10.1016/j.atmosres.2010.05.008 , 2010

Henneberger, J., Ramelli, F., Spirig, R., Omanovic, N., Miller, A. J., Fuchs, C., Zhang, H., Bühl, J., Hervo, M., Kanji, Z. A., Ohneiser, K., Radenz, M., Rösch, M., Seifert, P., and Lohmann, U.: Seeding of Supercooled Low Stratus Clouds with a UAV to Study Microphysical Ice Processes - An Introduction to the CLOUDLAB Project, Bulletin of the American Meteorological Society, -1, https://doi.org/10.1175/BAMS-D-22-0178.1 , 2023.715.

Ohata, S., Moteki, N., Mori, T., Koike, M., and Kondo, Y.: A Key Process Controlling the Wet Removal of Aerosols: New Observational Evidence, Scientific Reports, 6, 34 113, https://doi.org/10.1038/srep34113, 2016.

Seidel, D. J., Ao, C. O., and Li, K.: Estimating Climatological Planetary Boundary Layer Heights from Radiosonde Observations: Comparison of Methods and Uncertainty Analysis, Journal of Geophysical Research: Atmospheres, 115, https://doi.org/10.1029/2009JD013680, 2010.